# Organic matter degradation causes enrichment of organic pollutants in hadal sediments

Anna Sobek [1] ✉, Sebastian Abel[1], Hamed Sanei [2], Stefano Bonaglia [3], Zhe Li [1], Gisela Horlitz[1], Arka Rudra[2], Kazumasa Oguri[4,5] & Ronnie N. Glud [4,6,7]

Burial of persistent organic pollutants (POPs) such as polychlorinated biphenyls (PCBs) in deep-sea sediments contributes to 60% of their historical emissions. Yet, empirical data on their occurrence in the deep-ocean is scarce. Estimates of the deep-ocean POP sink are therefore uncertain. Hadal trenches, representing the deepest part of the ocean, are hotspots for organic carbon burial and decomposition. POPs favorably partition to organic carbon, making trenches likely significant sinks for contaminants. Here we show that PCBs occur in both hadal (7720–8085 m) and non-hadal (2560–4050 m) sediment in the Atacama Trench. PCB concentrations normalized to sediment dry weight were similar across sites while those normalized to sediment organic carbon increased exponentially as the inert organic carbon fraction of the sediment increased in degraded hadal sediments. We suggest that the unique deposition dynamics and elevated turnover of organic carbon in hadal trenches increase POP concentrations in the deepest places on Earth.

Man-made persistent organic pollutants (POPs) are found everywhere on Earth including the deep oceans[1–5]. These hazardous substances have physicochemical properties that make them persistent in the environment and undergo long-range transport[6]. POPs therefore have the potential to harm marine life far from their sources and long after their use was banned and primary emissions ceased. As an example, global emissions of the industrial chemicals polychlorinated biphenyls (PCBs) were significantly reduced when they were banned in the mid-1970s[7]. Yet, PCBs were recently demonstrated to threaten the reproduction and viability of more than 50% of the world's killer whale populations - almost five decades after use and emissions peaked[8]. PCBs are known to cause serious health effects in both humans and animals. They harm the reproductive and immune systems and are carcinogenic (The Stockholm Convention). PCBs bioaccumulate and as a result can reach harmful concentrations in consumers at the top of the food web, including humans.

Most POPs are poorly soluble in water and favorably partition to organic carbon in the water column. For instance, PCBs are concentrated ~1,000,000 times as they partition from water to phytoplankton[9]. The biological pump facilitates the vertical export of POPs sorbed to particles from the ocean surface to deep-water layers and ultimately to benthic deposition[10]. Vertical profiles of POPs in the water column of the Arctic Ocean showed increasing concentrations with depth as a result of transport with sinking particles in combination with long residence times of Arctic deep-water layers[1,11]. Other studies performed in the Atlantic and Indian Oceans down to 3000 m depth[3,12] concluded that ocean currents can play an important role for transporting POPs to deeper water layers. The deep ocean is considered a significant sink for POPs[10], and burial of PCBs in ocean sediments is estimated to correspond to 60% of cumulative emissions since the start of their production in the 1930s[2]. However, these assessments are based on a very limited number of measurements in

[1]Department of Environmental Science, Stockholm University, Stockholm, Sweden. [2]Lithospheric Organic Carbon (LOC) Group, Department of Geoscience, Aarhus University, Aarhus, Denmark. [3]Department of Marine Sciences, University of Gothenburg, Gothenburg, Sweden. [4]HADAL and Nordcee, Department of Biology, University of Southern Denmark, Odense, Denmark. [5]Research Institute for Global Change, Japan Agency for Marine-Earth Science and Technology, Yokosuka, Japan. [6]Danish Institute for Advanced Study (DIAS), University of Southern Denmark, Odense, Denmark. [7]Department of Ocean and Environmental Sciences, Tokyo University of Marine Science and Technology, Tokyo, Japan. ✉e-mail: anna.sobek@aces.su.se

the deep ocean and there is hardly any empirical data on the occurrence of POPs in sediments from the deepest part of the ocean.

Hadal trenches represent the deepest part of the global ocean stretching from 6 to 11 km of depth and covering an area of $3.44 \times 10^6 \text{ km}^2$, corresponding to about 1% of the ocean bed[13]. The trench systems act as collectors for organic material. This process is facilitated by tidal induced internal seiche[14] and downslope gravitational driven sediment displacement[15]. Apart from the more continuous downslope material transport, distinct sudden translocation of large amounts of previously deposited material, typically induced by earthquakes, may occur along the trench axis. Detailed investigation of $^{210}\text{Pb}_{ex}$ (gamma-ray spectrometry of excess $^{210}\text{Pb}$) profiles have shown that such "mass wasting" events contribute significantly to material deposition in the Atacama trench[15]. Hadal sediments harbor high abundance of microbial life that thrive at the extreme hydrostatic pressure[16–18], and have been found to support higher degradation rates of organic carbon as compared to adjacent abyssal sediment and rates often scale with values encountered at continental slopes and margins[19,20]. However, hadal carbon mineralization rates vary considerably within and between trench systems depending on i) surface ocean production[19], ii) local bathymetry and hydrographic conditions that may focus or winnow depositing material[14] and iii) the occurrence of infrequent mass wasting events translocating relict but also fresh labile organic material to the trench axis[19,21].

Data on the occurrence of POPs in hadal sediment are scarce. The two existing surveys in the peer-reviewed literature report either "non-detects"[4], or surprisingly overall high and diverse concentrations of POPs[22]. Further, the latter example includes unexplained relative contributions of individual PCB congeners[22]. This caveat is detected when comparing the relative contribution of single PCB congeners found in an environmental sample with its contribution in commercial PCB mixtures[23,24] (Aroclor) constituting the environmental source of PCBs. As an example, the maximum contribution of PCB#60 in Aroclor mixtures is <3%; yet, its relative contribution in the Mariana trench has been reported to be up to 65%[22]. Likewise, PCB#169 was reported in the same hadal setting to have a fairly high concentration (up to 2% of total PCB pool), although absent from Aroclor mixtures[23,24]. More studies on the occurrence of POPs in hadal sediment from the Mariana trench would be valuable to better understand earlier reports.

Two earlier studies demonstrate high concentrations of POPs in scavenging benthic amphipods collected from hadal sediments (Mariana, Mussau, New Britain and Kermadec trenches), with concentrations comparable to those found in contaminated areas (the seven indicator PCBs: 147–905 ng g$^{-1}$ dry weight in the Mariana and 18–43 ng g$^{-1}$ dry weight in the Kermadec trench[4,5] and 103–678 ng g$^{-1}$ lipid weight in the Mariana, and 7 to 65 ng g$^{-1}$ lipid weight in the Mussau and New Britain trenches[6]). No POPs were detected in sediment from the Mariana, Mussau and New Britain trenches by Cui et al[4], and sediment analyses were not performed in the study by Jamieson et al[5]. Recent discoveries have identified hadal trench regions to host benthic mercury accumulation rates 30–60 times higher than the deep ocean average[25,26], and elevated concentrations of arsenic[27]. Hence, the limited existing data calls for more research on the role of trench sediment as global sinks for contaminants.

Here we used highly-sensitive and selective tandem mass spectrometry to obtain datasets on PCB concentrations in sediment collected below 6000 m depth, and demonstrate concentrations of individual PCB congeners in the Atacama trench to be in the pg g$^{-1}$ dry weight (dwt) range. Our results suggest that the unique deposition dynamics in hadal trenches and elevated turnover of organic carbon contribute to increased POP sediment concentrations. We hypothesize this observation is a consequence of up-concentration of PCBs during organic carbon degradation, and possibly because of the increased sorption capacity of degraded organic matter.

## Results And Discussion

### Persistent organic pollutants found in sediment at around 8000 m depth in the Atacama trench

We analyzed PCBs from surficial sediments (0–10 cm) collected with push cores at five sites in the Atacama trench region at depths ranging from bathyal sites at 2500 m to hadal depths of more than 8000 m (Fig. 1; Table 1). The Atacama trench is formed by the subduction of the Nazca plate below the continental plate of South America. It extends about 4200 km off the cost of Peru and Chile, with a maximum depth of ca. 8000 m (23°36.79'S; 71°34.73'W[13]). The trench is located close to an intense upwelling region leading to high surface ocean productivity. Consequently, the trench is characterized by relatively high sediment accumulation rates with values along the trench axis ranging from 0.29 to 0.79 cm y$^{-1}$ and mass accumulation rates of organic carbon from 1.1 to 5.3 g C m$^{-2}$ y$^{-1}$[15]. The intense deposition of material is facilitated by down-slope focusing and seismic driven mass wasting events. Analysis of stable carbon isotopes suggest that the deposition of organic material of marine origin is supplemented by terrigenous sources potentially through run off in the south and windborne deposition in the North[28].

Five sediment layers, of 20 mm each, were analyzed from each site. PCBs are hydrophobic chemicals, with high tendency to sorb to organic carbon. The quality and quantity of organic carbon in the sediment samples therefore contributes to the understanding of the environmental fate of these substances. The sediment samples were analyzed for Total Organic Carbon (TOC) content and concentrations of labile protohydrocarbons (S1; mg HC g$^{-1}$ sediment) were measured to assess the degradability of the TOC (Supplementary Table 1). The proportion of the Inert Fraction of TOC was calculated. Sedimentation

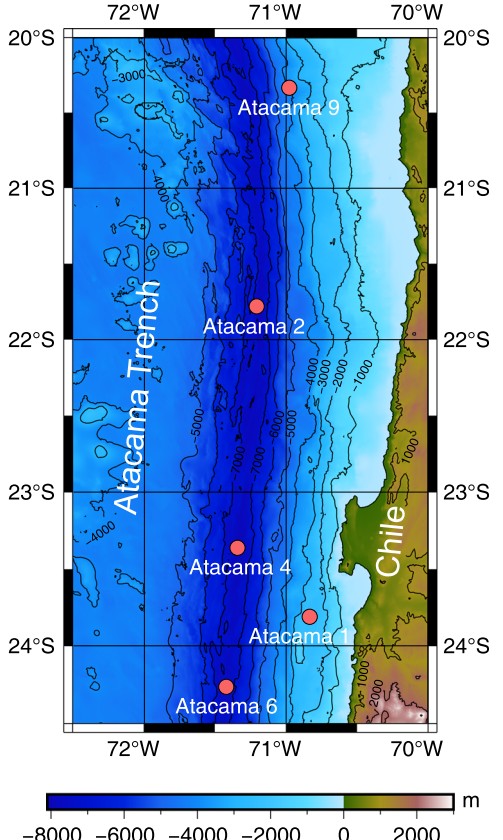

**Fig. 1 | Map of the Atacama trench area with sampling sites indicated.** The latitudinal distance between 20°S and 24.5°S is 500 km. The map is drawn with the GEBCO_2022 Grid dataset[48].

**Table 1 | Sampled sites in the Atacama trench area, with coordinates, water depth and Total Organic Carbon (TOC) content (%) and Inert fraction (%) in surface sediment (0–10 mm)**

| Site | Latitude | Longitude | Depth (m) | TOC[a] in surface sediment % | Inert Fraction[b] % | Earliest approximate year of deposition[c] 0–100 mm |
|---|---|---|---|---|---|---|
| Atacama 1 | 23° 48.72' S | 70° 50.04' W | 2560 | 2.31 | 71.0 | 1828–1883 |
| Atacama 2 | 21° 46.86' S | 71° 12.48' W | 7995 | 0.62 | 88.7 | 1953 |
| Atacama 4 | 23° 21.78' S | 71° 20.60' W | 8085 | 0.60 | 86.3 | 1917–1922 |
| Atacama 6 | 24° 15.96' S | 71° 25.38' W | 7720 | 0.68 | 86.8 | 1787–1796 |
| Atacama 9 | 20° 19. 97' S | 70° 58.70' W | 4050 | 0.96 | 80.5 | 1717 |

Sampling was done in 2018[15], PCBs were analyzed in five sediment layers of each 20 mm from each site. The earliest approximate year of deposition is based on $^{210}Pb_{ex}$
[a]TOC differs slightly from TOC data reported in Oguri et al[15]. due to analyses performed by different procedures, in two different laboratories and on parallel samples from different sediment cores. The data used in this study was generated in the same lab as the other OC parameters. The two data sets are not significantly different.
[b]The Inert Fraction is the percentage of the inert organic carbon (non-generative organic carbon, NGOC) in the TOC (NGOC/TOC×100). It represents the relative fraction of the highly refractory organic carbon that can only break down during intense oxidation processes, normally not available in the subsurface sediments or oxygen restricted water.
[c]Calculated based on sedimentation rates from Oguri et al.[15].

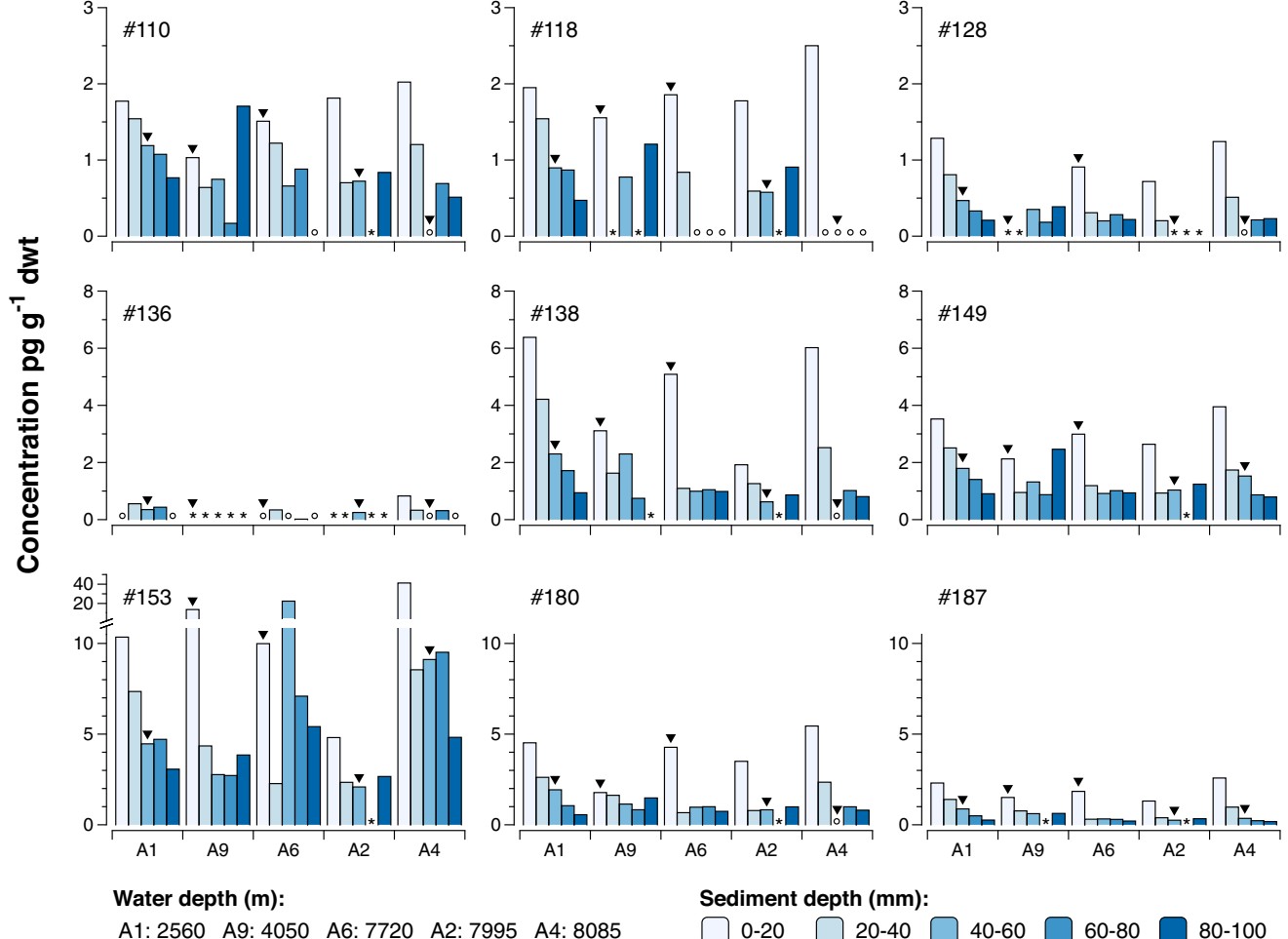

Water depth (m):
A1: 2560  A9: 4050  A6: 7720  A2: 7995  A4: 8085

Sediment depth (mm):
☐ 0-20  ☐ 20-40  ☐ 40-60  ☐ 60-80  ☐ 80-100

**Fig. 2 | Concentrations (given as pg contaminant per g of dry sediment) of nine polychlorinated biphenyl (PCB) congeners in the five upper sediment layers collected in the Atacama trench area (A1-A9).** Asterisks and circles indicate peaks cannot be integrated and concentrations below detection limit, respectively. Arrows indicate approximate sediment depth corresponding to time for peak global emissions of PCBs in the mid 1970s. Core mixing in surface sediment is presumably mediated by bioturbation and physical mixing.

rates per sediment sample and per site were estimated using gamma-ray spectrometry of excess $^{210}Pb$ ($^{210}Pb_{ex}$)[15]. This analytical method utilizes decay of $^{210}Pb_{ex}$ (half-life of 22.3 years) as a tracer for sediment mixing and deposition during the past ~120 years. Vertically constant values typically indicate mixing events induced either by seismic activity or bioturbation, while exponential decline in deeper sediment layers typically reflect stable and constant deposition rates[15]. The insight provides important information on pathways and chronology for deposited sediment layers. More specific details on material

deposition and organic carbon turnover in the Atacama trench region can be found elsewhere[15,19]. Briefly, benthic mineralization was dominated by aerobic mineralization and bottom water concentrations of $O_2$ and $NO_3^-$ were similar along the trench axis. Further, the $O_2$ penetration depth was generally similar across sites, ranging between 3.1 and 4.1 cm along the trench axis[19,29]. The analytical chemical method for detecting PCBs in sediment was optimized to minimize blank contamination, and involved analysis on a GC-MS/MS, using 1–2 qualifier ions for each PCB congener to increase accuracy and minimize

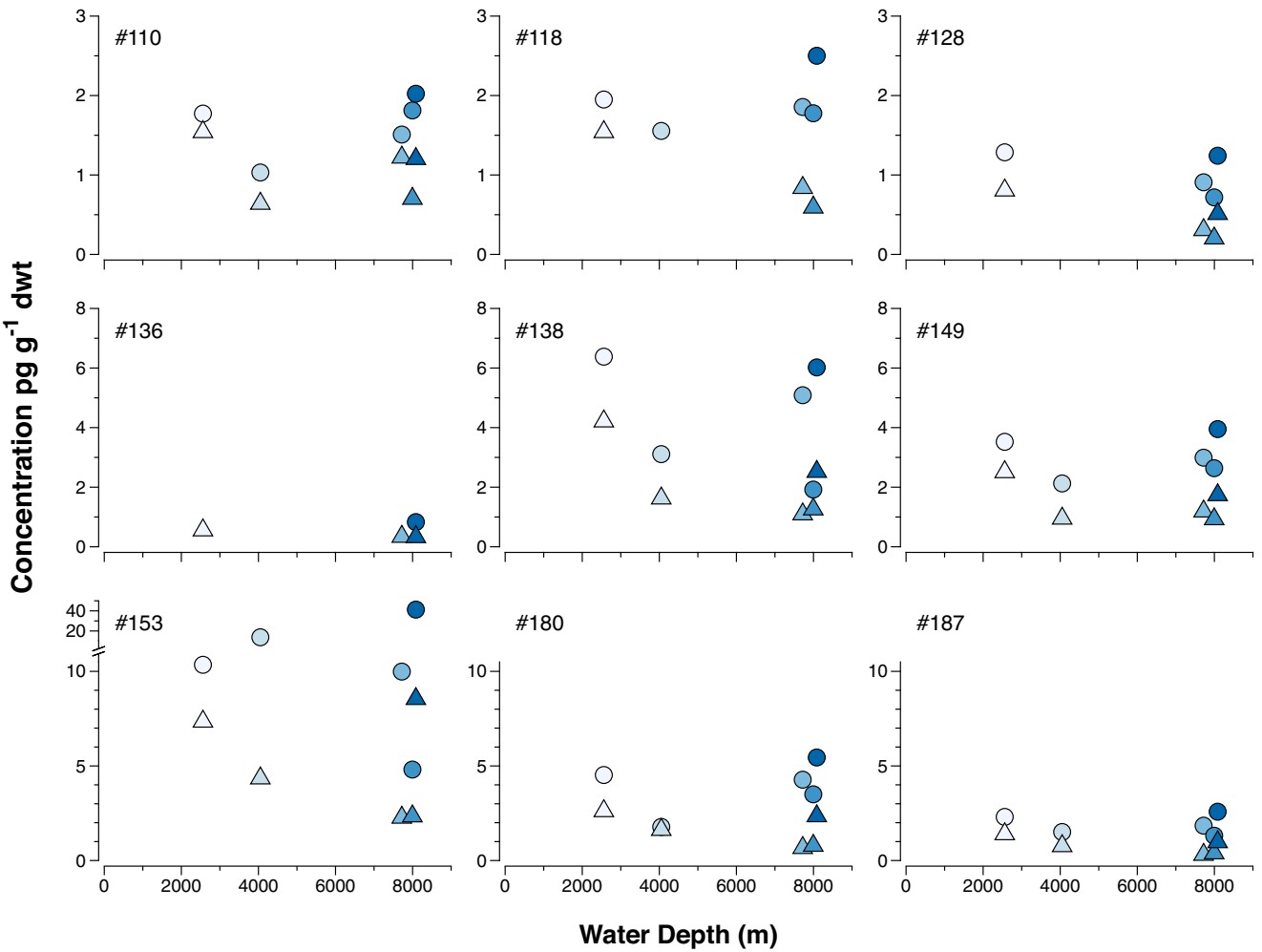

**Fig. 3 | Concentrations of nine polychlorinated biphenyl (PCB) congeners in pg g⁻¹ dwt in Atacama sediment (upper 40 mm) versus water depth.** Sediment concentrations in layer 0–20 mm are depicted by circles and in layer 20–40 mm by triangles. Colors from bright to dark indicate water depth from shallow to deep.

false positive detects. We found PCBs in sediment across all sites in all of the 50 sediment layers.

Observed concentrations of nine PCB congeners in surface sediment (upper 20 mm) ranged between 0.72 and 41.2 pg g⁻¹ dwt. Four of the seven indicator PCBs used by the International Council for the Exploration of the Sea (ICES) were found (#118, #138, #153, #180), and two of these (PCB#118 and PCB#180) environmentally ubiquitous PCBs are highlighted here for comparison with other studies (data on all congeners available in Fig. 2 and Supplementary Table 2). Concentrations of PCB#118 ranged between 1.5 and 2.5 pg g⁻¹ dwt and PCB#180 between 1.8 and 5.5 pg g⁻¹ dwt. PCB concentrations in the Atacama sediments are comparable to or lower than those reported earlier from the Mariana trench (PCB#118 0–13 pg g⁻¹ dwt; PCB#180 0–19.3 pg g⁻¹ dwt). Concentrations of several other PCB congeners reported from the Mariana trench were surprisingly high, and the authors of that study concluded that concentrations were "far higher than those recorded before in marine sediments from shallower depths"[22]. The PCB concentrations reported here are lower than those reported from surficial sediments collected from Arctic shelf seas (0 up to 50 mm; PCB#118 2–64 pg g⁻¹ dwt; PCB#180 1–41 pg g⁻¹ dwt)[30,31]. PCBs were mainly produced and used in the North, between 30°N and 60°N[7]. Emissions of PCBs therefore mainly occurred in the Northern Hemisphere. These compounds are semi-volatile and can be transported through the atmosphere, by rivers and ocean currents. Through the so called grasshopping and cold condensation effects, PCBs are transported to the Arctic, and to a high extent kept there due to the

influence of low temperatures on the mobility of these chemicals[32]. Therefore, concentrations of PCBs can be expected to be higher in the Arctic compared to in remote areas in the Southern Hemisphere. Deng et al.[33] analyzed PCBs in sediment collected in the Antarctica (154–3240 m depth) and the South China Sea (1380–4000 m depth). PCB#180 was not included in that study, but PCB#118 (co-eluting with PCB#106 and #108) had reported concentrations of 160–280 pg g⁻¹ dwt in the Antarctica and 20–40 pg g⁻¹ dwt in the South China Sea, thus 1–2 orders of magnitude higher than our observations in the Atacama trench. The detected PCB concentrations in the Atacama trench are 300–1500 fold lower when compared to those measured from heavily impacted marine settings like the Baltic Sea[34,35], which contrasts to previous observations of PCB concentrations in hadal amphipods[4,5]. Yet, the findings of persistent, toxic and bioaccumulating man-made chemicals in some of the deepest and presumed most isolated regions of the global ocean is noteworthy.

### No peak concentrations of PCBs in Atacama sediment cores
For all five sampling sites the highest concentrations of PCBs were observed in the surface sediment (top 20 mm), and concentration generally decreased with sediment depth (Fig. 2). The surface sediment at the two shallowest sites (Atacama 1, Atacama 9) may be affected by bioturbation (see Supplementary Fig. 1 for photo of sediment core)[15]. While larger bioturbating infauna in hadal sediment were rare as evidenced by the distinct layering in Supplementary Fig. 1, epifauna such as scavenging amphipods and holothurians could still

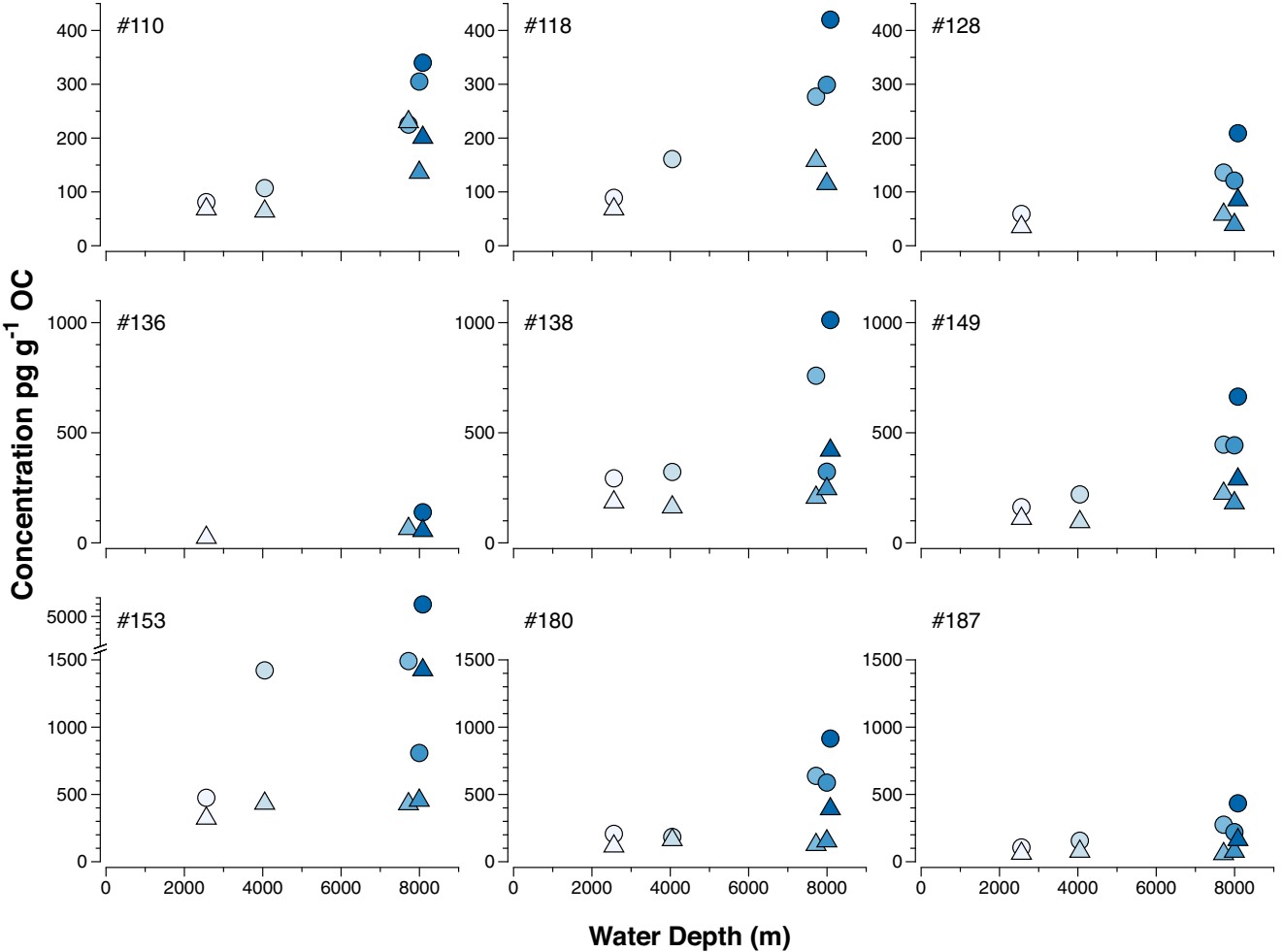

**Fig. 4 | Concentrations of nine polychlorinated biphenyl (PCB) congeners in pg g$^{-1}$ organic carbon (OC) in Atacama sediment (upper 40 mm) versus water depth.** Sediment concentrations in layer 0–20 mm are depicted by circles and in layer 20–40 mm by triangles. Colors from bright to dark indicate water depth from shallow to deep.

contribute to mixing of the surface layers[15]. In addition, hadal environments have complex deposition dynamics with ongoing downslope material transport towards the trench axis and infrequent mass wasting events that translocate large amounts of material from the slopes to the trench axis[21,36]. Based on $^{210}Pb_{ex}$ dating, the upper 100 mm of material should have been deposited approximately within the past 65–300 years (Table 1).

The detection of PCBs in the deepest analyzed sediment layers at several sites is a sign of sediment mixing, as PCBs cannot be present in sediment deposited decades before the onset of industrial production of PCBs (in the 1930s). Our findings of generally higher concentrations in the upper-most sediment layers may therefore need to be interpreted with caution. Yet, our observations of increasing concentrations towards surface sediment layers contrast to the commonly reported peak concentrations of POPs in sediment layers deposited during the 1980s–1990s[34,37] (Fig. 2). For PCBs, global emissions were significantly reduced in the mid-1970's[7], followed by a stop of global production of PCBs in the 1990s. If confirmed in future studies, the observation of generally increasing PCB concentrations towards the sediment surface may indicate that sediment concentrations have not yet peaked in this remote area, with important implications for dispersal time for POP deposition in trenches.

In the shallow and contaminated Baltic Sea (water depth ≤459 m), PCB sediment concentrations in coastal areas peaked in the late 1970s, while offshore areas demonstrated peak

concentrations about 15 years later[34], presumably due to internal transport processes. Data on PCBs in dated sediment cores from the southern hemisphere are scarce. Available data from shallow freshwater systems in Mexico[38] and Tanzania[39], however demonstrate time trends of PCBs that are different from those reported from the north. The Tanzanian core demonstrated a peak concentration of PCBs in sediment in the early 2000s, followed by a significant decline, and in recent years a steep concentration increase. Potential explanations include secondary sources, such as release from electronic waste containing PCBs, to contribute to the increasing PCB concentrations in the last decade[39]. In the two Mexican sediment cores, time trends were similar to what we observed in the Atacama. No peak concentrations were found since the 1970's, but PCB concentrations were increasing since the 1990's in one of the cores, and during the last decade in the other core[38]. In the hadal Atacama sites (>6000 m depth), sedimentation rates were low (0.43–1.53 mm year$^{-1}$;[15]), meaning that on average 20 mm sediment layer corresponded to 27–69 years. It is therefore possible that peak concentrations occurring in the late 1990's to early 2000s, were not detected in the subsurface Atacama sediment layers due to low time resolution of the 20 mm samples. It is also possible, that concentrations have not yet peaked in this remote area, or that mass waste events or bioturbation, disguise the effect of decreasing PCB emissions in the trench area. Future studies are needed to further investigate the

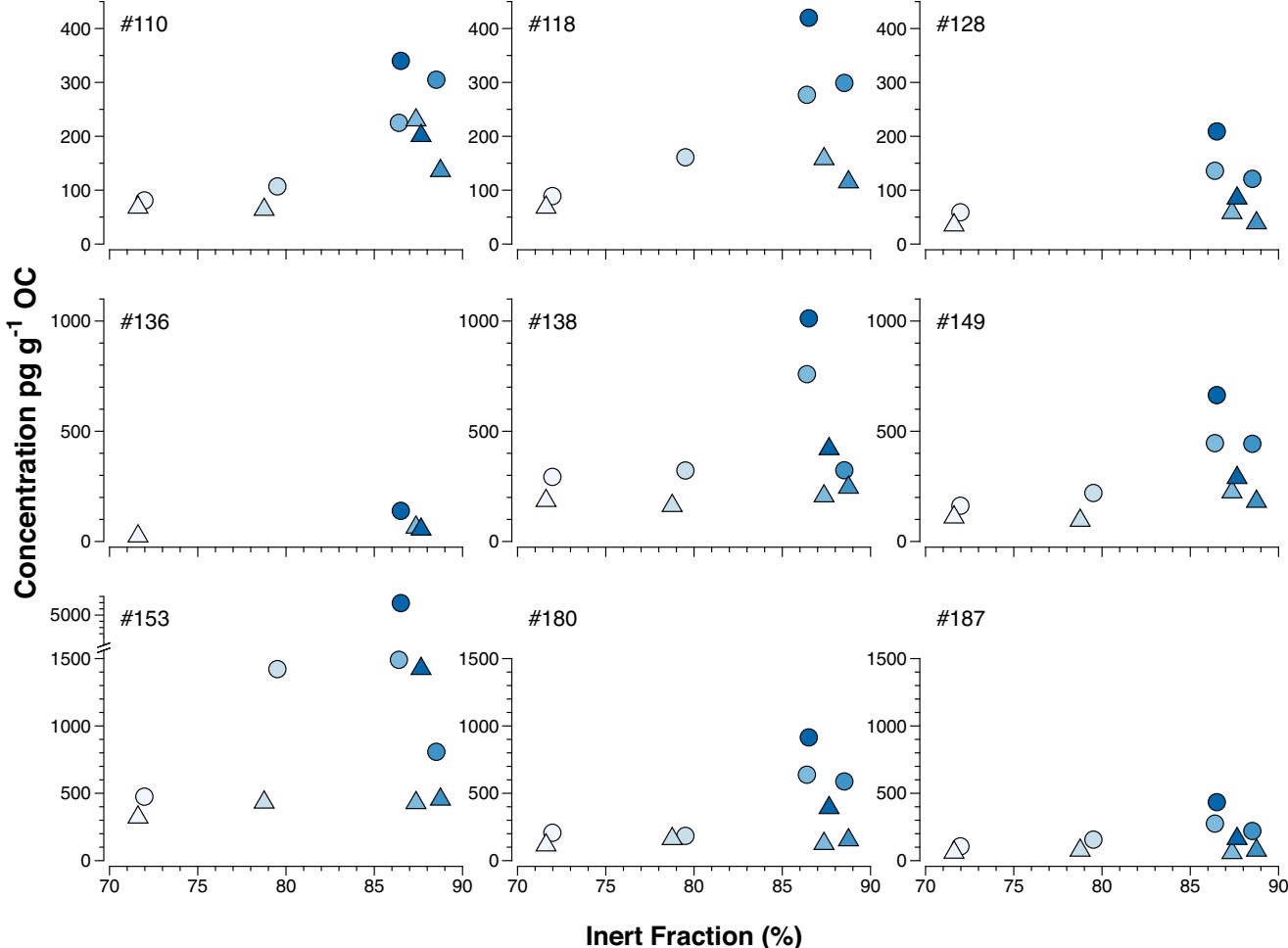

**Fig. 5 | Concentrations of nine polychlorinated biphenyl (PCB) congeners in pg g⁻¹ organic carbon (OC) in Atacama sediment (upper 40 mm) versus the inert fraction (%; non-generative organic carbon fraction/total organic carbon;** NGOC/TOC). Sediment concentrations in layer 0–20 mm are depicted by circles and in layer 20–40 mm by triangles. Colors from bright to dark indicate water depth from shallow to deep.

concentrations of PCBs and other legacy POPs in sediment cores from the deep sea. If POPs in deep sea sediment have not yet reached peak concentrations, it poses questions about the time scale of global transport of POPs.

**Organic matter turnover as a driver for PCB sediment concentrations**

Sediment concentrations of PCBs (normalized to sediment dry weight) in the two upper sediment layers (0–20 mm and 20–40 mm) were similar across hadal (Atacama 2, 4, 6) and non-hadal (Atacama 1, 9) sites (Fig. 3).

PCBs are hydrophobic chemicals (log $K_{ow}$ ~4–8), with low water solubility and they preferably partition to organic matter in the environment. Consequently, environmental concentrations of PCBs are commonly expressed on an organic-carbon basis (pg g⁻¹ OC). Normalizing the Atacama PCB concentrations to sediment organic carbon content reveals a pattern where the non-hadal sites tend to have the lowest PCB concentrations (Fig. 4).

Atacama 1, sampled at 2560 m depth on the slope of the continental margin, represents a different setting compared to sites within the trench, with material focusing and complex deposition dynamics[19]. Hadal trenches, including the Atacama, have relatively high turnover rates of organic carbon[19]. As PCBs selectively partition to organic matter, degradation of organic carbon leads to elevated PCB concentrations on an organic carbon basis. It is also likely that the bulk organic carbon quality as sorbent for PCBs changes as it degrades in

the trench. While a fraction of the labile organic carbon is preserved along the trench axis[15,40], the inert fraction of the surface sediment organic matter generally increases with the water depth at the targeted sites. There was a marked difference in PCB concentrations of the hadal and non-hadal sites. Concentrations of PCBs per mass organic carbon plotted versus the inert organic carbon fraction show exponential relationships (Fig. 5), indicating that the concentration of PCBs in the unit of organic carbon increases exponentially as the proportion of organic carbon in sediments become more inert/refractory.

Further, the proportion of the labile protohydrocarbons (S1) in sediment organic carbon shows an inverse relationship with PCB content per organic carbon for the study sites (Fig. 6). This corroborates the above finding that PCBs appear to be affiliated with the highly refractory fraction of organic carbon in the study sediments. PCBs and other organic contaminants with an aromatic backbone sorb strongly to condensed organic matter such as soot and charcoal[41]. This effect is particularly strong for e.g., PAHs which have a planar configuration, but has also been observed for PCBs and other less planar compounds[42,43]. Hence, the formation and accumulation of the most degraded, refractory, inert organic matter fraction in the studied trench sediment, could lead to higher sorption capacity for POPs. In line with this, Hawthorne et al[44] observed stronger sorption of PCBs to weathered compared to less weathered sediment using a database of more than 1900 sediment samples. Krauss and Wilcke[45] studied occurrence and sorption of PCBs and PAHs to soil fractions of increasing density. They found decreasing concentrations of both

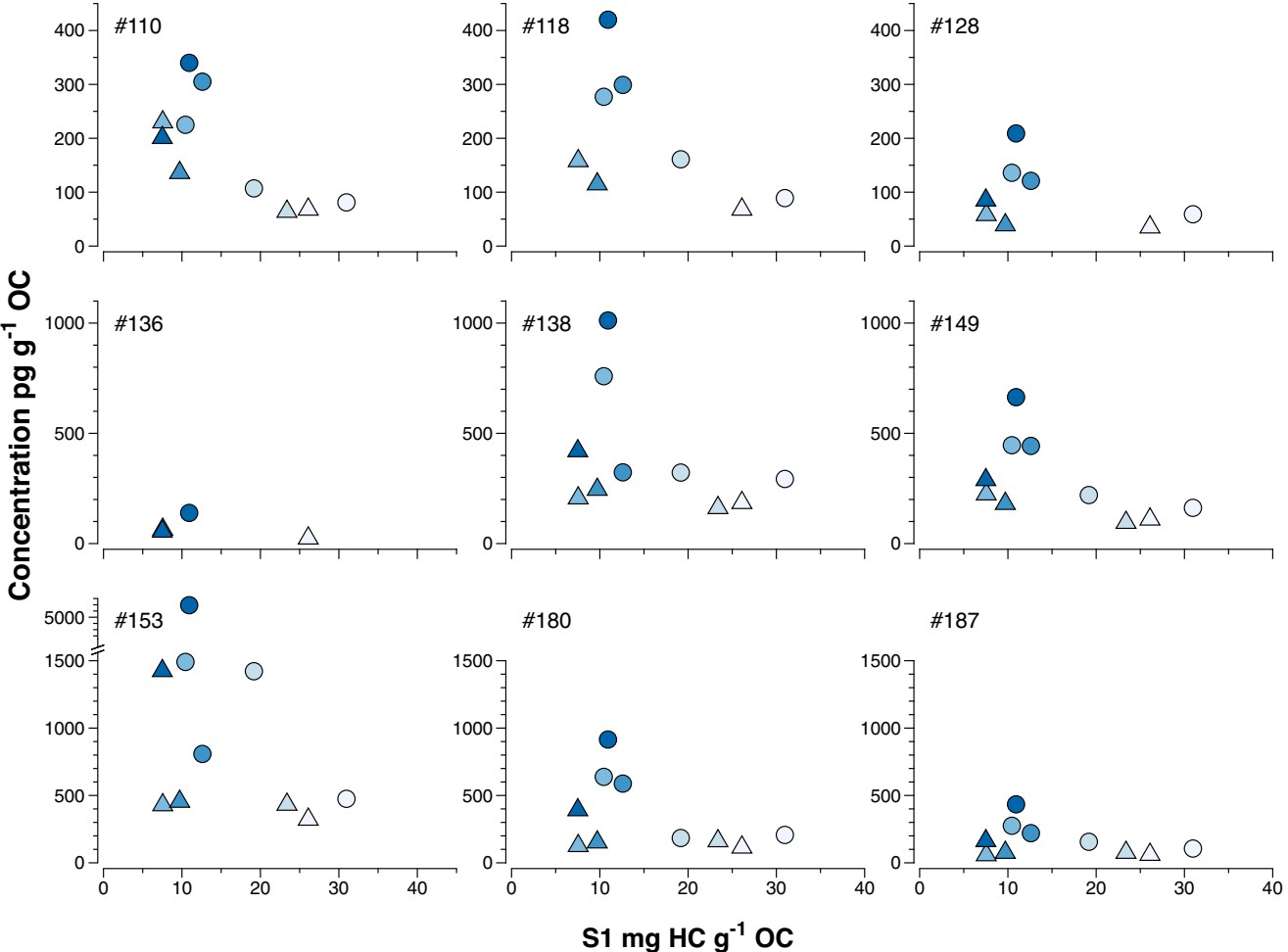

**Fig. 6 | Inverse relationship between concentrations of nine polychlorinated biphenyl (PCB) congeners in pg g$^{-1}$ organic carbon (OC) and those of the labile protohydrocarbons (HC) (S1 mg HC g$^{-1}$ OC) in the Atacama surface sediment (upper 40 mm).** Sediment concentrations in layer 0–20 mm are depicted by circles and in layer 20–40 mm by triangles. Colors from bright to dark indicate water depth from shallow to deep.

PAHs and PCBs with increasing soil density. Further, OC-normalized PAH concentrations were significantly higher in the heavy, more degraded, fraction compared to lighter fractions. This effect was more pronounced for heavier PAHs, while no such trend was observed for PCBs. Thus, the distribution of PCBs in the different soil fractions were driven by organic matter quantity and not quality, while the heavier PAHs were also affected by OC quality.

In the Atacama sediment samples, the range in physicochemical properties among the measured and quality assured PCBs was too narrow to establish any effects of organic matter quality on distribution within the trench. This limitation also prevented further investigations of the possibility of anaerobic dechlorination of highly chlorinated PCB congeners in hadal sediment samples[46]. Yet, we hypothesize that our results reflect a combination of PCBs being more persistent than the sediment organic carbon leading to an enrichment of PCBs as organic matter degrades on the trench slope and focus along the trench axis[19,20], and that the degraded organic carbon has a higher capacity to sorb POPs compared to more labile carbon[44].

## Methods
### Sampling
Sediment cores from Atacama were collected during the cruise *R/V Sonne* (SO261, 2018) using a multiple core sampler[47]. Sediment cores were sliced onboard and sediment samples were stored frozen in clean blue-cap bottles with aluminum foil secured between the glass and the plastic cap until further processing in the lab. Thawing and sub-sampling for contaminant analysis was done in a clean fume hood pre-tested for potential background contamination, and samples were stored in pre-cleaned glass jars. $^{210}Pb_{ex}$ concentration in the sediments were measured with gamma-ray analysis as described by Oguri et al.[15].

### Sediment organic carbon
Geochemical measurement of the sediment organic carbon was conducted in the Lithospheric Organic Carbon (LOC) lab, Department of Geoscience, Aarhus University. The method involves programmed temperature, open anhydrous pyrolysis, and combustion of ~50 mg dry bulk sediments using the HAWK Pyrolysis and TOC analyzers (Wildcat Technologies, USA). Concentrations of the labile proto-hydrocarbons (S1; mg HC g$^{-1}$ sediment) were measured using a FID detector at iso-temperature of 300 °C. The remaining concentrations of the hydrocarbons and the oxygen containing organic carbon (CO and $CO_2$) released during pyrolysis at ramping heat (25 °C/min) of up to 650 °C would sum up to the total generative organic carbon (GOC wt%) fraction. The remaining organic carbon was combusted in the oxidation oven to temperature of 800 °C, which is attributed to the remaining refractory, non-generative organic carbon fraction (NGOC wt%) or often regarded as the "inert organic carbon" in this paper. Total organic carbon (TOC wt%) is sum of the GOC and NGOC. The analytical accuracy and precision were monitored using the WT2 standard (Wildcat Technologies, USA).

## PCB analyses

Sediment samples were dried by mixing with sodium sulfate ($Na_2SO_4$) before extraction. For extraction, 3–16 g dry sediment was weighed into Accelerated Solvent Extraction (ASE) cells and 20 μL of a PCB surrogate standard solution in iso-octane (100 pg μL$^{-1}$ of $^{13}$C-labeled congeners #28, #52, #101, #118, #138, #153 and #180) was added. The extraction cells were filled up with diatomaceous earth to improve extraction efficiency. The extraction was carried out using acetone:n-hexane (1:1, V/V) at 100 °C for two cycles yielding ca. 60 mL extract from each cell, which was then evaporated to near-dryness and exchanged to pure n-hexane. A final two-step cleanup process consisted of a sulfuric acid and activated copper treatment to remove organic matrix components and elemental sulfur. Prior to GC-MS/MS analysis, a recovery standard solution containing ($^{13}$C-PCBs #111) was added to each sample.

Due to the low expected concentrations, a large volume injection method was used. In brief, 10 μL sample were injected into a programmable temperature vaporizing (PTV) inlet, where the excess solvent was evaporated at low temperatures prior to injection onto the column. Each congener was analyzed with a minimum of two transitions in the MS/MS detector (Supplementary Table 3). The data was quantified based on the transition ion with the strongest signal of each analyte, using an internal standard method. One to two secondary transitions were used to qualify each analyte. The labeled PCB standards were assigned to each analyte based on chlorination degree (Supplementary Table 4).

## QA/QC

Blank samples were run in parallel with each batch of sediment samples, to account for potential contamination during drying, extraction, cleanup and analysis. Congener-specific method detection limits were calculated as the average of batch-specific amounts in the blank samples plus three standard deviations. For samples without measurable levels in blanks, the method detection limits (Supplementary Table 5) were determined based on signal-to-noise-ratio (peak height 6×/10× above noise). Internal standard recoveries were calculated using the ratio of responses of each individual surrogate congener and the recovery standard ($^{13}$C-PCBs#111). A reference mix consisting of surrogate and recovery standards was used for comparison (see Supplementary Table 4 for individual recoveries of each congener). Our method validation demonstrated low recoveries/precision for lighter PCBs. These congeners (<PCB#110) were therefore removed from the dataset.

## Data availability

All data generated or analyzed during this study are included in this published article and its supplementary information file.

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

## Acknowledgements

Samples were obtained during cruises by the *R/V Sonne*, cruise SO261 awarded to Frank Wenzöfer, Matthias Zabel and Ronnie N. Glud. We thank captain and crew for excellent support to obtain the samples. Financial support was provided by HADES-ERC Advanced grant "Benthic diagenesis and microbiology of hadal trenches" #669947, the Danish National Research Foundation grant DNRF145 (Danish Center for Hadal Research). S.B. was supported by the Swedish Research Council Formas (grant number: 2017-01513). Additional financial supports are from the Department of Environmental Science at Stockholm University.

## Author contributions

A.S., S.B., and R.N.G. designed the project. R.N.G. organized the expe-dition and performed the sediment sampling. A.S., S.A., Z.L., and G.H. planned and performed the chemical analyses. H.S., R.N.G., and A.R. planned and performed the organic carbon analyses. K.O and R.N.G sampled for and realized the investigations on sediment chronology. All authors discussed the results and their implications. A.S. wrote the manuscript with contributions from all the authors.

## Funding

## Competing interests

The authors declare no competing interests.
