## [Peer Review File · Nature Communications]

Organic matter degradation causes enrichment of organic pollutants in hadal sedimentsReviewer #1 (Remarks to the Author):

In this study, the authors detected PCBs in sediment cores collected from the Atacama Trench and found organic carbon deposition played a role in the distribution of PCBs in hadal trenches. Previous studies have detected PCBs in the deep sea, including in hadal trenches. The novelty of the present study is they detected the PCBs in Atacama Trench.

Considering the scarcity of studies on PCBs in deep sea, particularly hadal trenches, and the toxicity of PCBs, this study is of relevance to the field. However, it is not clear why only these seven were analyzed considering more could be analyzed using the GC-MS/MS approach. The method used can easily detect at least 40 PCB congeners, 20 organochlorine pesticides, and 10 PBDEs.

The work used reporting concentration as a fraction of organic carbon as an indicator of the deposition dynamics within the hadal trenches. As rightfully shown in the Introduction, organic carbon deposition in hadal trenches is complex and there are numerous routes of entry of PCBs into hadal trenches, which makes normalizing concentration with organic carbon too simplistic. Just reporting the concentrations as a fraction of organic carbon and then make deposition dynamics conclusions might not be adequate.

The data presentation made it difficult to follow the work. For example, one figure is on PCB153, the other is on PCB138, and the next is on PCB110, 118 and 138. It is not clear why only these congeners were presented in the manuscript when it is possible to present all 7 using appropriate data analysis tools.

Specific comments

L25-27: the two sentences are contradictory; they give empirical statement of fact (75%) and yet the next sentence says there is scarce evidence. The 75% came from Sobek and Gustafsson (2014) who estimated the amount of PCBs stored in Arctic Ocean shelf sediments, specifically.

L29: with burial (see L25, L28, etc) do you mean deposition or sink? You may want to use the appropriate term to avoid confusion.

L29-30: there are several studies on POPs in hadal sediments and there is a review article published in 2021 that synthesized all the studies.

L36: The abstract does not mention any findings or commentary on deposition dynamics in hadal trenches apart from this sentence.

L59: the study cited estimated cumulative storage rather than emission. The only comment on emission the study made was on how insignificant climate change could contribute to the emissions from shelf sediment.

L60-62: as previously mentioned there are studies that have been done on the deepest parts of the ocean, commenting on the limitations of those studies might be a more expedient way of presenting the knowledge gap you seek to address (as you did in L74-77).

L74-77: it is not clear whether the discussion is focused on PCBs only or all POPs that have been detected in hadal zones

L77-83: it is not clear what the insinuation of 'unexplained relative contribution of individual PCB congeners' is. Are the results from previous studies questionable because of the deviations from industrial congener ratios or is it possible there are some physicochemical or biogeochemical processes in hadal trenches or during the transport to hadal trenches that these congeners found it higher ratios are recalcitrant towards? Either way, there is need for clarification.

L135-137: what about the remaining 5 indicator PCBs?

L141-142: why are these findings noteworthy? Why are the concentrations in Atacama trench which is close to a continental shelf where there is significant anthropogenic activity lower than in the Arctic?

Reviewer #2 (Remarks to the Author):

The paper by Sobek et al., reports measurements on persistent organic pollutants and their speciation in Atacama trench. The authors collected surficial sediments (up to 10 cm depth) from hadal and non-hadal sites in Atacama, and performed a suite of analyses including GC-MS/MS and radiotracing/radiochronologie to detect/speciate PCBs, and assess sediment mobilization/redistribution. Likewise, they performed TOC analyses that would help them to further interpret their PCB data.

Hadal trenches are geographically remote and challenging oceanic settings, and any information obtained from these ecosystems is precious. The novelty of the paper relies on the fact that human-derived PCBs are identified in Atacama trench. This leaves open the possibility that similar fate exists also for organic pollutants in other hadal trenches. While evidence of accumulation of heavy metals has been reported in hadal trenches, presence of organic pollutants at such deep depths has not previously reported accurately. This reviewer is in favor of the authors to publish their work in Nat Comm. Nevertheless, high-tier journals are also addressing to readers with different expertise. So studies published in those journals need to include also a broader context of information. This will make them more appealing and interesting to the broader audience.

I provide below comments, edits and suggestions. Because line and page numbering are missing, I copied-pasted sentences or phrases from the text and then provided feedback. A is abbreviation for authors and R for reviewer.

Abstract

A1: "This makes trenches likely hotspots also for POP burial, as hydrophobic POPs favorably partition to organic carbon."

R1: True, as long as POPs are not already utilized/uptaken/absorbed from particles/microorganisms in the water column prior of reaching to deeper depths or sediments. You mention this in your intro. Please rephrase accordingly.

A2: "Concentrations normalized to sediment dry weight (pg/g)"

R2: Correct pg/g to pg g⁻¹

A3: "However, PCB concentrations"

R3: Correct to PCB sediment concentrations

Introduction

A1: "Yet, PCBs were recently demonstrated to threaten the reproduction and viability of more than 50% of the world's killer whale populations - almost five decades after use and emissions peaked⁹."

R1: The effect of PCB accumulation is also detrimental to other species (e.g. toxic/carcinogenic in humans). Also, microorganisms reported to degrade PCBs are usually those that have also the potential to form biofilms. So, it is beneficial for the reader if you highlight better the overall severity of PCBs, on organisms.

A2: "Data on the occurrence of POPs in hadal zones is scarce. The very few existing reports in the peer-reviewed literature on POPs in hadal sediment either report non-detects⁵, or surprisingly high but variable levels with an unexplained relative contribution of individual PCB congeners²¹. The latter can be assessed by comparing the relative contribution of single PCB congeners found in an environmental sample with its contribution in commercial PCB mixtures^{22,23} (Aroclor) constituting the environmental source of PCBs."

R2: Please rephrase as follows: "Data on the occurrence of POPs in hadal zones is scarce. The handful existing surveys in the peer-reviewed literature report either "non-detects"⁵, or surprisingly high and diverse concentrations of POPs in hadal sediments. Further they include unexplained relative contribution of individual PCB congeners²¹. This caveat is detected when comparing...."

A3: "For example, the maximum contribution of PCB#60 in Aroclor mixtures is less than 3%, yet a relative contribution of up to 65% of this substance was reported in the Mariana Trench²¹. Another example is PCB#169, which was reported in fairly high concentration representing up to 2% of total mixture) concentration, although it is absent in Aroclor mixtures^{22,23}."

R3: Rephrase as follows: "As an example, the maximum contribution of PCB#60 in Aroclor mixtures is < 3%; yet, its relative contribution in the Mariana Trench has been reported to be up to 65%²¹. Likewise, PCB#169 was reported in the same hadal setting to have a fairly high concentration (up to 2% of total PCB pool), although absent from Aroclor mixtures^{22,23}."

Also show consistency with the usage of #.

A4: "Two earlier reports"

R4: Rephrase as "Two earlier studies"

A5: "to those found in contaminated areas" (R5: please provide here concentration range xxx-zzz ng g⁻¹)

A6: "Recent discoveries have identified hadal trench regions to host benthic mercury accumulation rates 30-60 times higher than the deep ocean average^{24,25}

R6: Data on mercury and arsenic concentrations exist also for Challenger Deep sediments, and show different accumulation rates between slopes (~6-8 km) vs. bottom-axis sites (~11km depth), and vs. non hadal reference sites (please see Zhou et al., 2022).

A7: "...the first datasets on PCB concentrations in sediment collected below 6000 m depth, and demonstrate concentrations of individual PCB congeners in the Atacama trench to be in the pg g⁻¹ dry weight (dwt) range."

R7: Please briefly introduce Atacama trench to the reader. Also, justify why you choose these sites shown in Figure 1. Are there any similarities or differences between your chosen stations (e.g., does Atacama 1 receives terrigenous input due to proximity to land), or selection was based solely on bathymetry? Also, it would be good if the map contained a bar scale for the reader to estimate distances between sites.

Results and Discussion

Persistent Organic Pollutants found in sediment at around 8000 m depth in the Atacama trench

A1: "We analyzed PCBs in sediment"

R1: Rephrase as follows: We analyzed PCBs from surficial sediments (0-10 cm) collected with push cores at five...

A2: "Sediment accumulation rates were derived on profiles of excess ²¹⁰Pb (²¹⁰Pb_{ex}) in the respective sediment layer²⁶"

R2: Rephrase: Sedimentation rates per sediment sample and per site were estimated using gamma-ray spectrometry of ²¹⁰Pb (²¹⁰Pb_{ex})²⁶. This analytical method utilizes Pb radionuclides as sediment tracers and profiles the ²¹⁰Pb distribution to examine XXX.

A3: Table 1.

R3: You mention TOC for the first time in the table legend. This need to be introduced in the text since you utilized these values to interpret your data in last section.

Also, provide in the text in a way that is easy for the reader to understand the similarities with Oguri et al., (sites and TOC measurements). Explain if your TOCs are, or are not, statistically different from those reported by Oguri et al., "Slightly differ" and "average 15%" differences can be in the marginal error due to the different methodological approaches, and the overall heterogenous nature of the sediments. But the way phrased is vague (average of 15% more, less, significant or not).

Do the authors have TOC values for deeper sediment interiors? If yes, please report. If not, please explain in the text why you report only for the 0-10 mm (e.g., challenging samples? Low TOC recovery?). (see also comment in the last section of Results Discussion). Does your study measured other environmental data (inorganic nutrients, T, O₂ etc?)

Finally, although you present the ²¹⁰Pb_{ex} values you comment them briefly (if at all) in the text. The ²¹⁰Pb_{ex} shows quite recent events in terms of geological time. Can you elaborate briefly in the text?

A4: "Observed PCB concentrations in surface sediment (upper 20 mm) varied"

R4: You write earlier in the text: "Five sediment layers, of 20 mm each, were analyzed from each site". Figure 3 show 0-10, and then 10-30 mm 30-50 mm (up to 90 mm with 20 mm sediment intervals). So, this (upper 20 mm) needs to be 0-20 mm, and probably explained better. Also, replace "varied" with ranged.

If not mistaken, you overall have 3 or 4 out of 7 PCB congeners used by ICES. So, it might be beneficial to give the general overview that you identified 6 PCB congeners with concentrations ranging from ~ 1.5 pg g⁻¹ to 14(?) pg g⁻¹. And then continue by explaining that you detected three/four out of the 7 marker PCBs used by the International Council for the Exploration of the Sea (ICES) to determine degree of PCB contamination in the sea (ref). And then focus on PCB110 PCB180 and explain why you chose these 2 specifically.

A5: These concentrations correspond to, or are lower than, the reported concentrations in Arctic Ocean shelf surface (0-10 mm; 0-20 mm or 0-50 mm) sediments^{28,29}, while being several orders of magnitude lower than concentrations found in areas known to be heavily impacted by anthropogenic activities, such as the Baltic Sea^{30,31}. Yet the findings in some of the deepest and presumed most isolated regions of the global ocean is noteworthy.

R5: This is a long sentence; I have edited it and broke it in two parts. "These concentrations correspond to, or are lower than those reported from surficial sediments collected from Arctic shelf seas and Arctic interior basins, (0 up to 50 mm; XX -ZZ ng g⁻¹)^{28,29}. Nonetheless, the detected PCBs are XX-fold lower when compared to those measured from heavily impacted marine settings like the Baltic Sea^{30,31}. Yet, the findings in some of the deepest and presumed most isolated regions of the global ocean is noteworthy.

I understand the comparison to other marine environments, especially if they are remote (regardless if hadal or not), but the Baltic? Anything compared to Baltic is less..The authors have extensively worked in PCBs. Nonetheless, this reviewer believes that including more relative comparisons with whatever deep-sea data are available, even if data are sketchy (e.g., Dasgupta et al 2018), needs to be made. The potential caveats of such comparisons can be pointed out in the text in a sentence.

(e.g., <https://www.sciencedirect.com/science/article/pii/S0025326X15000648>)
"Yet, the findings in some of the deepest and presumed most isolated regions of the global ocean is noteworthy."

Well, this is your hot finding. You need to put it upfront already from your abstract. For this reviewer the fact that you present the first data on PCBs is important, but not as important as that PCBs are found in geographically remote and isolated ocean areas, like trenches.

A6: Figure 2.

R6: Better to reorganize Figure 2 with depth (shallow to deeper stations) instead of site numbers (1-9). Also increase scale bar to accommodate Atacama 4 (unless you have a larger scale and is not clearly shown on my copy). If levels on At4 are high, and visually undermine the other concentrations, then either use // and "break" the At 4 153, or make it a multi-panel figure. Also, I would suggest the authors to search for statistically significant differences between congeners and between stations, and comment appropriately.

A7: "The surface sediment at the two shallowest sites (Atacama 1, Atacama 9) is most certainly affected by bioturbation."

R7: The way written is confusing. You write: "is most certainly affected by bioturbation". The legend of Photo S1 from the whole core says: "Distinct laminated structures in deeper layers suggest absence of bioturbation by infauna and occasional deposition events." Please, elaborate. Also, if you find bioturbation in the surface it would be beneficial to point it with an arrow or something in the photo.

Now, If bioturbation is prominent why it should be restricted only in the shallow sites? Evidence of organisms that bioturbate exists at deeper depths of other hadal trenches (Puerto Rico trench), albeit at lower abundancies. So someone could argue that bioturbation can occur at deeper hadal depths but at lower rates due to low abundancies of bioturbators.

A8: mass wasting events

R8: Define for the reader (e.g., XX).

A9: "slopes to the trench axis^{20,32}"

R9: This is true. Nonetheless, differences of organic matter occur also between the different sediment interiors, independent of slope or bottom axis sites, and could be due

to different available nutrient pools (e.g. NO_x, NH₄) and presence/absence of O₂. All these parameters control the microbial community structure, and thus how organic matter is utilized/disproportionate. So please elaborate.

A10: The detection of PCBs in the deepest analyzed sediment layers at several sites is a sign of sediment mixing.

R10: Vaguely written. Also, this is evident from what?

A11: with some care

R11: Consider "with caution" or "overinterpretation" as terms.

A12: "Yet, our observations of increasing concentrations towards surface sediment layers contrast to the commonly reported peak concentrations of POPs in sediment layers deposited during the 1980s-1990s^{30,33}, following upon significantly reduced global emissions of PCBs in the mid-1970's and a stop of global production of PCBs in the 1990s (Figure 3)."

R12: The reader needs clearly to understand why this is important. The data sets you are referring to are from organic-rich coastal sediments, and sediments from offshore marine settings and harbors. These areas have a history of pollution and accumulation of PCBs. So, rephrase to indicate the strength of your contradictory finding, and provide potential explanation. Also, this is a long sentence. Break it in 2.

A13: Figure 3.

R13: Explain why you picked PCB138 (be consistent with #). Replace: "Missing data means the concentration was below detection limit" with "Missing data indicate PCB#138 concentrations below the detection limit."

Also, Atacama 2 and Atacama 4 are X km apart, they are collected from approximately same water depths, have same levels of TOC, they present their peaks at 50 mm, their 210Pbex profiles are similar (max difference is 36 yrs which in chronological ages can be considered marginal); yet one site has measurable PCB#138 while the other does not. Now, if you compare these findings with those from the third deeper site (Atacama 6) you see differences on the layer where you find the peak, and also on the 210Pbex profile. Can you discuss more these differences?

A14: In the shallow (average 52 m, maximum 459 m) and contaminated Baltic Sea

R14: In the shallow and contaminated Baltic Sea (water depth ≤ 459 m),

A15: "Secondary sources such as release from"

R15: Potential explanations include secondary sources....

A16: occurring in the late 1990's to early 2000s, were not detected in the subsurface Atacama sediment layers.

R16: Can you put this is under a general context? E.g., 1) changes in legislation that helped in PCB reduction, 2) geographically remoteness of Atacama, 3) you sampled deep-sea and hadal depths so depositions/accumulations of PCBs, can be less profound because of 1) and 2)? Just a suggestion.

Organic matter turnover as a driver for PCB sediment concentrations

R1: This section is nicely written. Some overall comments to be considered:

1. You are referring to organic matter degradation by microbes. Right? If yes, then please comment briefly (sentence or two) about energy processes/microbial communities of Atacama.

2. You make a valid comparison between PCBs and PAHs. Depending on the degree of aromaticity, presence of oxygen, redox conditions etc, PAHs can be degraded by microbes. Hadal studies have shown potential for microbial hydrocarbon degradation (e.g., Mariana trench). So, does the suggested release of PCBs upon organic matter degradation, could also imply that fraction of these PCBs can be utilized by hadal microbes? Microbial diversity in trenches is quite large and novel species exist with unknown metabolic potentials. You might want to point in your text something similar.

3. Would you consider an overall schematic representation of what you suggest?

A2: "Sediment concentrations of PCBs"

R2: Rephrase to: Sediment concentrations of three marker PCBs

A3: expressed on an organic-carbon basis (pg g⁻¹ OC).

R3: For this normalization did you use your TOC data? How did you deal with the sediment depths below 10 mm?

REVIEWER COMMENTS

Reviewer #1 (Remarks to the Author):

In this study, the authors detected PCBs in sediment cores collected from the Atacama Trench and found organic carbon deposition played a role in the distribution of PCBs in hadal trenches. Previous studies have detected PCBs in the deep sea, including in hadal trenches. The novelty of the present study is they detected the PCBs in Atacama Trench.

RESPONSE: It is correct that there are some, although few, reports on the occurrence of PCBs in hadal trenches. We want to underline that – to the best of our knowledge – there are only two previous studies that intended to measure PCBs in hadal *sediment*. Cui et al. did not detect any PCBs in sediment (although they reported high levels in hadal amphipods). Dasgupta et al. reported occurrence of PCBs in hadal sediment, however for some congeners in surprisingly high concentrations for some congeners (see further comments below).

Considering the scarcity of studies on PCBs in deep sea, particularly hadal trenches, and the toxicity of PCBs, this study is of relevance to the field. However, it is not clear why only these seven were analyzed considering more could be analyzed using the GC-MS/MS approach. The method used can easily detect at least 40 PCB congeners, 20 organochlorine pesticides, and 10 PBDEs.

RESPONSE:

We analyzed 21 congeners in the sediment samples, but as described in the manuscript, due to low recoveries of the lighter congeners, several of them were removed from the study (<PCB#110). PCB#170 and #209 were analyzed but not detected in any of the samples, this information was added to Table S2 in Supplementary Information. We present data on nine PCB congeners (#110, #118, #128, #136, #138, #149, #153, #180, #187), of which 4 belong to the indicator PCBs (#118, #138, #153, #180). It is possible that additional congeners could have been detected, but the included ones belong to commonly found PCBs in the environment and are thus representative of environmental pollution and offer comparability to other studies. PCB concentrations in hadal sediment are very low, as demonstrated by Cui et al., who did not detect any PCBs in hadal sediment samples.

The reviewer asks for data on PBDEs and organochlorine pesticides (OCPs). Based on results from other samples from less remote areas in Northern Europe, we did not add standards of PBDEs or OCPs to the samples as it was unlikely that we would detect them with the instrumentation we had available at that time (we did not have access to a GC-MS/MS initially).

The work used reporting concentration as a fraction of organic carbon as an indicator of the deposition dynamics within the hadal trenches. As rightfully shown in the Introduction, organic carbon deposition in hadal trenches is complex and there are numerous routes of entry of PCBs into hadal trenches, which makes normalizing concentration with organic carbon too simplistic. Just reporting the concentrations as a fraction of organic carbon and then make deposition dynamics conclusions might not be adequate.

RESPONSE:

Presenting PCB concentrations in environmental samples (including soil, sediment, plants, phytoplankton) normalized to organic carbon is a standard procedure (see e.g., Schwarzenbach et al., Environmental Organic Chemistry, Wiley & Sons). The reason is that there is overwhelming

evidence that hydrophobic molecules, such as PCBs, partition to organic matter (see e.g., Nizzetto et al., 2009). The normalization is needed to show possible variability in environmental concentrations driven by other factors than organic carbon content. In our study (start line 249-319), we show that not only organic carbon content but also its turnover in the trench, may be important for the sediment PCB concentration. We demonstrate this by illustrating how the organic carbon-normalized concentration correlates to the degradability of the sediment organic carbon (Figures 3-6). We thus argue for maintaining the presentation as it is.

The data presentation made it difficult to follow the work. For example, one figure is on PCB153, the other is on PCB138, and the next is on PCB110, 118 and 138. It is not clear why only these congeners were presented in the manuscript when it is possible to present all 7 using appropriate data analysis tools.

RESPONSE: The figures have been revised according to suggestions by the reviewer. All analyzed congeners (9) are now included.

SPECIFIC COMMENTS

L25-27: the two sentences are contradictory; they give empirical statement of fact (75%) and yet the next sentence says there is scarce evidence. The 75% came from Sobek and Gustafsson (2014) who estimated the amount of PCBs stored in Arctic Ocean shelf sediments, specifically.

RESPONSE:

The first statement comes from a model study of PCBs in the global ocean (Wagner et al., 2019). The model demonstrates that deep sea sediments constitute a significant sink for PCBs. However, the model is based on very few data points and although they propose a percentage estimate for global sediment removal of PCBs, this estimate is uncertain and there is need for more data.

In response to the reviewer, we revised the first part of the abstract, which now reads (line 22-24):

“Burial of persistent organic pollutants (POPs) in deep-sea sediments may correspond to up to 60% of historical emissions. Yet, empirical data on the occurrence of POPs in the deep ocean is scarce. Estimates of the magnitude of the deep ocean POP sink are therefore uncertain.”

L29: with burial (see L25, L28, etc) do you mean deposition or sink? You may want to use the appropriate term to avoid confusion.

RESPONSE:

In the scientific literature, burial in the deep sea has been described as one of two sinks for POPs in the environment, with the other one being atmospheric degradation through the OH radical (Dachs et al., 2002). To clarify, we have adjusted the wording (line 26-28):

“This makes trenches likely significant sinks for POPs, as hydrophobic POPs favorably partition to organic carbon, which may eventually transport to the sediment.”

L29-30: there are several studies on POPs in hadal sediments and there is a review article published in 2021 that synthesized all the studies.

RESPONSE:

To the best of our knowledge, we have included all available data on earlier reports of POPs in hadal sediment. The reviewer refers to a review article from 2021, but does not provide any details about authors or journal. We assume he/she is referring to the one by Du et al., 2021, named “Geology, environment, and life in the deepest part of the world’s oceans” published in The Innovation (doi.org/10.1016/j.xinn.2021.100109). In this review, there is a paragraph about POPs in hadal sediment, which is based on the same articles we refer to in our manuscript. No other reports are mentioned. The review article highlights the reports on POPs in amphipods by Jamieson et al and Cui et al, which we also discuss in our manuscript. The authors of the review article further bring up the data on high concentrations of POPs in hadal sediment reported by Dasgupta et al, a study which we comment in our manuscript. No other studies on POPs in hadal sediment were reported in the review article. We therefore find it fair to say that we present among the first data on the occurrence of POPs in hadal sediment.

L36: The abstract does not mention any findings or commentary on deposition dynamics in hadal trenches apart from this sentence.

RESPONSE:

This is correct. The focus of this study was the concentrations of PCBs in Atacama hadal sediment and how they can be better understood by organic carbon degradation processes and deposition. The deposition dynamics of hadal environments have been discussed in several papers including three recent manuscripts from our group. However, to accommodate the critique, the essence of deposition pathways in hadal settings is now summarized in two sentences and appropriate references are included (line 67-70):

“This process is facilitated by tidal induced internal seiche and downslope gravitational driven sediment displacement. Apart from the more continuous downslope material transport, distinct sudden translocation of large amounts of previously deposited material, typically induced by earthquakes, may occur along the trench axis.”

L59: the study cited estimated cumulative storage rather than emission. The only comment on emission the study made was on how insignificant climate change could contribute to the emissions from shelf sediment.

RESPONSE:

We did not completely understand this comment. The study by Wagner et al., 2019, which we refer to in line 62, demonstrates through a global model, that burial in marine sediments “accounts for cumulative removal of approximately 75% of the PCBs **released to the atmosphere since 1930**”. Hence, the reviewer is correct in that it refers to cumulative storage, but also cumulative emissions. No changes to the manuscript have been made.

L60-62: as previously mentioned there are studies that have been done on the deepest parts of the ocean, commenting on the limitations of those studies might be a more expedient way of presenting the knowledge gap you seek to address (as you did in L74-77).

RESPONSE:

In this sentence, we conclude that assessments on burial of POPs in the deep ocean are uncertain as they are “based on a very limited number of measurements and there is hardly any empirical data on the occurrence of POPs in sediments from the deepest part of the ocean”. As discussed in one of

our responses to the reviewer above, there are currently two earlier studies which aimed to analyze POPs in hadal sediment. These are, i) the study by Cui et al, which did not detect any POPs in hadal sediment, and ii) the study by Dasgupta et al, which detected high and diverse concentrations of POPs in hadal sediment. We discuss both these studies in the lines following the sentence, starting on line 81:

“Data on the occurrence of POPs in hadal sediment is scarce. The two existing surveys in the peer-reviewed literature report either “non-detects”, or surprisingly overall high and diverse concentrations of POPs. Further, the latter example includes unexplained relative contributions of individual PCB congeners.”

Further, we have added and improved comparisons of our findings with reports of PCBs in other (deep sea) locations, including the data by Dasgupta et al., starting on line 172:

“PCB concentrations in the Atacama sediments are comparable to or lower than those reported earlier from the Mariana trench (PCB#118 0-13 pg g⁻¹ dwt; PCB#180 0-19.3 pg g⁻¹ dwt). Concentrations of several other PCB congeners reported from the Mariana Trench were surprisingly high, and the authors of that study concluded that concentrations were “far higher than those recorded before in marine sediments from shallower depths”. The PCB concentrations reported here are lower than those reported from surficial sediments collected from Arctic shelf seas, (0 up to 50 mm; PCB#118 2-64 pg g⁻¹ dwt; PCB#180 1-41 pg g⁻¹ dwt). PCBs were mainly produced and used in the North, between 30°N and 60°N. Emissions of PCBs therefore mainly occurred in the Northern Hemisphere. These compounds are semi-volatile and can be transported through the atmosphere, by rivers and ocean currents. Through the so called grasshopping and cold condensation effects, PCBs are transported to the Arctic, and to a high extent kept there due to the influence of low temperatures on the mobility of these chemicals. Therefore, concentrations of PCBs can be expected to be higher in the Arctic compared to in remote areas in the Southern Hemisphere. Deng et al analyzed PCBs in sediment collected in the Antarctica (154-3240 m depth) and the South China Sea (1380-4000 m depth). PCB#180 was not included in that study, but PCB#118 (co-eluting with PCB#106 and #108) had reported concentrations of 160-280 pg g⁻¹ dwt in the Antarctica and 20-40 pg g⁻¹ dwt in the South China Sea, thus 1-2 orders of magnitude higher than our observations in the Atacama Trench.”

L74-77: it is not clear whether the discussion is focused on PCBs only or all POPs that have been detected in hadal zones

RESPONSE:

We refer to POPs in general, but our specific example is about PCBs. For clarity, we revised the manuscript to clarify. The sentences now read (starting on line 81):

“Data on the occurrence of POPs in hadal sediment is scarce. The two existing surveys in the peer-reviewed literature report either “non-detects”, or surprisingly overall high and diverse concentrations of POPs. Further, the latter example includes unexplained relative contributions of individual PCB congeners.”

L77-83: it is not clear what the insinuation of ‘unexplained relative contribution of individual PCB congeners’ is. Are the results from previous studies questionable because of the deviations from industrial congener ratios or is it possible there are some physicochemical or biogeochemical processes in hadal trenches or during the transport to hadal trenches that these congeners found it higher ratios are recalcitrant towards? Either way, there is need for clarification.

RESPONSE:

The study by Dasgupta et al presents high PCB concentrations in the Mariana Trench; concentrations that cannot be understood based on comparisons with other studies, or even the proximity to sources at land. The presented PCB concentrations by Dasgupta et al also reveal an unexplained relative contribution of individual PCB congeners in the samples, as we mention in our manuscript. There might be natural – presently unknown – explanations to the concentrations presented in Dasgupta et al, and more data from the Mariana Trench would be valuable to help understand this better. In response to the reviewer, we have revised the text to clarify. The text now reads (line 90-92):

“More studies on the occurrence of POPs in hadal sediment from the Mariana trench would be valuable to better understand earlier reports.”

For transparency, we added comparisons of our measured concentrations to the ones by Dasgupta et al (line 172-176):

“PCB concentrations in the Atacama sediments are comparable to or lower than those reported earlier from the Mariana trench (PCB#118 0-13 pg g⁻¹ dwt; PCB#180 0-19.3 pg g⁻¹ dwt). Concentrations of several other PCB congeners reported from the Mariana Trench were surprisingly high, and the authors of that study concluded that concentrations were “far higher than those recorded before in marine sediments from shallower depths.””

L135-137: what about the remaining 5 indicator PCBs?**RESPONSE:**

We have revised this paragraph to clarify. In the earlier version of the manuscript, we chose to exemplify the variation in concentrations for two congeners. In the revised manuscript, we clarify that concentrations of the other analyzed congeners can be found in the Supplementary Information (see text starting on line 166):

“Observed concentrations of nine PCB congeners in surface sediment (upper 20 mm) ranged between 0.72 and 41.2 pg g⁻¹ dwt. Four of the seven indicator PCBs used by the International Council for the Exploration of the Sea (ICES) were found (#118, #138, #153, #180), and two of these (PCB#118 and PCB#180) environmentally ubiquitous PCBs are highlighted here for comparison with other studies (data on all congeners available in Figure 2 and Table S2). Concentrations of PCB#118 ranged between 1.5 and 2.5 pg g⁻¹ dwt and PCB#180 between 1.8 and 5.5 pg g⁻¹ dwt.”

L141-142: why are these findings noteworthy? Why are the concentrations in Atacama trench which is close to a continental shelf where there is significant anthropogenic activity lower than in the Arctic?**RESPONSE:**

We believe these findings are of interest as it is unique data. The hadal environment is very remote, and data that demonstrates the occurrence of anthropogenic pollution in this extreme environment is noteworthy, particularly given (1) the scarce availability of earlier data from hadal areas; (2) until very recently trench environments were thought to be pristine; (3) PCBs are extremely persistent and can easily bioaccumulate in sediment infauna and spread in the food web.

We added the following text to this paragraph, in response to this reviewer, and to motivate our statement (starting on line 179):

“PCBs were mainly produced and used in the North, between 30°N and 60°N⁷. Emissions of PCBs therefore mainly occurred in the Northern Hemisphere. These compounds are semi-volatile and can be transported through the atmosphere, by rivers and ocean currents. Through the so called grasshopping and cold condensation effects, PCBs are transported to the Arctic, and to a high extent kept there due to the influence of low temperatures on these chemicals’ mobility. Therefore, concentrations of PCBs can be expected to be higher in the Arctic compared to in remote areas in the Southern Hemisphere.”

In addition, we revised the statement about the importance of our study (line 192-194):

“Yet, the findings of persistent, toxic and bioaccumulating man-made chemicals in some of the deepest and presumed most isolated regions of the global ocean is noteworthy.”

Reviewer #2 (Remarks to the Author):

The paper by Sobek et al., reports measurements on persistent organic pollutants and their speciation in Atacama trench. The authors collected surficial sediments (up to 10 cm depth) from hadal and non-hadal sites in Atacama, and performed a suite of analyses including GC-MS/MS and radiotracing/radiochronologie to detect/speciate PCBs, and assess sediment mobilization/redistribution. Likewise, they performed TOC analyses that would help them to further interpret their PCB data.

Hadal trenches are geographically remote and challenging oceanic settings, and any information obtained from these ecosystems is precious. The novelty of the paper relies on the fact that human-derived PCBs are identified in Atacama trench. This leaves open the possibility that similar fate exists also for organic pollutants in other hadal trenches. While evidence of accumulation of heavy metals has been reported in hadal trenches, presence of organic pollutants at such deep depths has not previously reported accurately. This reviewer is in favor of the authors to publish their work in Nat Comm. Nevertheless, high-tier journals are also addressing to readers with different expertise. So studies published in those journals need to include also a broader context of information. This will make them more appealing and interesting to the broader audience. I provide below comments, edits and suggestions. Because line and page numbering are missing, I copied-pasted sentences or phrases from the text and then provided feedback. A is abbreviation for authors and R for reviewer.

ABSTRACT

A1: "This makes trenches likely hotspots also for POP burial, as hydrophobic POPs favorably partition to organic carbon."

R1: True, as long as POPs are not already utilized/uptaken/absorbed from particles/microorganisms in the water column prior of reaching to deeper depths or sediments. You mention this in your intro. Please rephrase accordingly.

RESPONSE:

We revised the sentence to (line 26-28):

"This makes trenches likely significant sinks for POPs, as hydrophobic POPs favorably partition to organic carbon, which may eventually transport to the sediment."

A2: "Concentrations normalized to sediment dry weight (pg/g)"

R2: Correct pg/g to pg g⁻¹

RESPONSE:

OK

A3: "However, PCB concentrations"

R3: Correct to PCB sediment concentrations

RESPONSE:

OK

INTRODUCTION

A1: "Yet, PCBs were recently demonstrated to threaten the reproduction and viability of more than 50% of the world's killer whale populations - almost five decades after use and emissions peaked9."

R1: The effect of PCB accumulation is also detrimental to other species (e.g. toxic/carcinogenic in humans). Also, microorganisms reported to degrade PCBs are usually those that have also the potential to form biofilms. So, it is beneficial for the reader if you highlight better the overall

severity of PCBs, on organisms.

RESPONSE:

In response to this comment, we added the following explanatory text (line 48-51):

“PCBs are known to cause serious health effects in both humans and animals. They harm the reproductive and immune systems and are carcinogenic (The Stockholm Convention). PCBs bioaccumulate and as a result can reach harmful concentrations in consumers at the top of the food web, including humans.”

A2: “Data on the occurrence of POPs in hadal zones is scarce. The very few existing reports in the peer-reviewed literature on POPs in hadal sediment either report non-detects⁵, or surprisingly high but variable levels with an unexplained relative contribution of individual PCB congeners²¹. The latter can be assessed by comparing the relative contribution of single PCB congeners found in an environmental sample with its contribution in commercial PCB mixtures^{22,23} (Aroclor) constituting the environmental source of PCBs.”

R2: Please rephrase as follows: “Data on the occurrence of POPs in hadal zones is scarce. The handful existing surveys in the peer-reviewed literature report either “non-detects”⁵, or surprisingly high and diverse concentrations of POPs in hadal sediments. Further they include unexplained relative contribution of individual PCB congeners²¹. This caveat is detected when comparing....”

RESPONSE:

We changed the text according to the suggestion by the reviewer, with some minor edits. It now reads (starting on line 81):

“Data on the occurrence of POPs in hadal sediment are scarce. The two existing surveys in the peer-reviewed literature report either “non-detects”, or surprisingly high and diverse concentrations of POPs. Further, the latter example includes unexplained relative contributions of individual PCB congeners. This caveat is detected when comparing...”

A3: “For example, the maximum contribution of PCB#60 in Aroclor mixtures is less than 3%, yet a relative contribution of up to 65% of this substance was reported in the Mariana Trench²¹. Another example is PCB#169, which was reported in fairly high concentration representing up to 2% of total mixture) concentration, although it is absent in Aroclor mixtures^{22,23}.

R3: Rephrase as follows: “As an example, the maximum contribution of PCB#60 in Aroclor mixtures is < 3%; yet, its relative contribution in the Mariana Trench has been reported to be up to 65%²¹. Likewise, PCB#169 was reported in the same hadal setting to have a fairly high concentration (up to 2% of total PCB pool), although absent from Aroclor mixtures^{22,23}.”
Also show consistency with the usage of #.

RESPONSE:

The change was made (line 87-90) and the manuscript is revised to be consistent with the usage of #.

A4: “Two earlier reports”

R4: Rephrase as “Two earlier studies”

RESPONSE:

OK

A5: “to those found in contaminated areas” (R5: please provide here concentration range xxx-zzz ng g-1)

RESPONSE:

We provided concentration ranges reported in these earlier studies. The sentence now reads (line 94-98):

“...with concentrations comparable to those found in contaminated areas (the seven indicator PCBs: 147-905 ng g⁻¹ dry weight in the Mariana and 18-43 ng g⁻¹ dry weight in the Kermadec trench and 103-678 ng g⁻¹ lipid weight in the Mariana, and 7 to 65 ng g⁻¹ lipid weight in the Mussau and New Britain trenches).”

A6: “Recent discoveries have identified hadal trench regions to host benthic mercury accumulation rates 30-60 times higher than the deep ocean average^{24,25}

R6: Data on mercury and arsenic concentrations exist also for Challenger Deep sediments, and show different accumulation rates between slopes (~6-8 km) vs. bottom-axis sites (~11km depth), and vs. non hadal reference sites (please see Zhou et al., 2022).

RESPONSE: We could not find data on mercury in the paper suggested by the reviewer, but we revised the sentence to include arsenic (line 100-102):

“Recent discoveries have identified hadal trench regions to host benthic mercury accumulation rates 30-60 times higher than the deep ocean average, and elevated concentrations of arsenic.”

A7: “...the first datasets on PCB concentrations in sediment collected below 6000 m depth, and demonstrate concentrations of individual PCB congeners in the Atacama trench to be in the pg g-1 dry weight (dwt) range.”

R7: Please briefly introduce Atacama trench to the reader. Also, justify why you choose these sites shown in Figure 1. Are there any similarities or differences between your chosen stations (e.g., does Atacama 1 receives terrigenous input due to proximity to land), or selection was based solely on bathymetry? Also, it would be good if the map contained a bar scale for the reader to estimate distances between sites.

RESPONSE: It is problematic to include a scale bar on Figure 1, as the map stretches across many degrees of latitude - the distance at the top and bottom of the map therefore differs (as it does on most 2D map of Globus). However, we have now included the length of the trench in the text so that it should provide some kind of scale for the reader.

We added the following information about the Atacama Trench, to clarify why this site was selected (first paragraph of the Results and Discussion, starting from line 117):

“The Atacama Trench is formed by the subduction of the Nazca plate below the continental plate of South America. It extends about 4200 km off the coast of Peru and Chile, with a maximum depth of ca. 8000 m (23°36.79' S 71°34.73' W). The trench is located close to an intense upwelling region leading to high surface ocean productivity. Consequently, the trench is characterized by relatively high sediment accumulation rates with values along the trench axis ranging from 0.29 to 0.79 cm y⁻¹ and mass accumulation rates of organic carbon from 1.1 to 5.3 g C m⁻² y^{-1,15}). The intense deposition is facilitated by down-slope focusing and seismic driven mass wasting events. Analysis of stable

carbon isotopes suggest that the deposition of organic material of marine origin is supplemented by terrigenous sources potentially through run off in the south and windborne deposition in the North.”

RESULTS AND DISCUSSION

Persistent Organic Pollutants found in sediment at around 8000 m depth in the Atacama trench

A1: “We analyzed PCBs in sediment”

R1: Rephrase as follows: We analyzed PCBs from surficial sediments (0-10 cm) collected with push cores at five...

RESPONSE:

The change was made.

A2: “Sediment accumulation rates were derived on profiles of excess ^{210}Pb ($^{210}\text{Pb}_{\text{ex}}$) in the respective sediment layer²⁶”

R2: Rephrase: Sedimentation rates per sediment sample and per site were estimated using gamma-ray spectrometry of ^{210}Pb ($^{210}\text{Pb}_{\text{ex}}$)²⁶. This analytical method utilizes Pb radionucleosides as sediment tracers and profiles the ^{210}Pb distribution to examine XXX.

RESPONSE:

The change was made.

A3: Table 1.

R3: You mention TOC for the first time in the table legend. This need to be introduced in the text since you utilized these values to interpret your data in last section.

RESPONSE:

We thank the reviewer for the comment, and agree it should be better explained. The following text was added to the paragraph (starting on line 128):

“PCBs are hydrophobic chemicals, with high tendency to sorb to organic carbon. The quality and quantity of organic carbon in the sediment samples therefore contributes to the understanding of the environmental fate of these substances. The sediment samples were analyzed for Total Organic Carbon (TOC) content and concentrations of labile protohydrocarbons (S1; mg HC g⁻¹ sediment) were measured to assess the degradability of the TOC. The proportion of the Inert Fraction of TOC was calculated. “

Also, provide in the text in a way that is easy for the reader to understand the similarities with Oguri et al., (sites and TOC measurements).

RESPONSE: We believe this is not necessary, as the focus of our study is the occurrence of PCBs in the Atacama Trench. To help in the interpretation of the PCB data, we use organic carbon dynamics, but as the TOC data from Oguri et al and our study are comparable (although not identical), we do not want to dilute our message with a long discussion about this. No changes were made to the manuscript.

Explain if your TOCs are, or are not, statistically different from those reported by Oguri et al., “Slightly differ” and “average 15%” differences can be in the marginal error due to the different methodological approaches, and the overall heterogenous nature of the sediments. But the way phrased is vague (average of 15% more, less, significant or not).

RESPONSE: We agree this became unnecessarily complicated. We simplified the text, which now reads (below Table 1):

“¹TOC differs slightly from TOC data reported in Oguri et al. due to analyses performed by different procedures, in two different laboratories and on parallel samples from different sediment cores. The data used in this study was generated in the same lab as the other OC parameters. The two data sets are not significantly different.”

Do the authors have TOC values for deeper sediment interiors? If yes, please report. If not, please explain in the text why you report only for the 0-10 mm (e.g., challenging samples? Low TOC recovery?). (see also comment in the last section of Results Discussion). Does your study measured other environmental data (inorganic nutrients, T, O₂ etc?)

RESPONSE: The samples for this study were taken along with a larger investigation on material transport and organic carbon turnover in the Atacama Trench (Oguri et al., 2022; Glud et al., 2021). Full profiles of additional parameters are available in Oguri et al 2022. This information was added to the revised manuscript (line 140-145):

“More specific details on material deposition and organic carbon turnover in the Atacama Trench region can be found elsewhere. Briefly, benthic mineralization was dominated by aerobic mineralization and bottom water concentrations of O₂ and NO₃⁻ were similar along the trench axis. Further, the O₂ penetration depth was generally similar across sites, ranging between 3.1-4.1 cm along the trench axis.”

Further, in response to the reviewer, we added data on TOC, S1 and Inert Fraction for all sediment samples (0-10 cm) to Table S1, Supplementary Information.

Finally, although you present the ²¹⁰Pb_{ex} values you comment them briefly (if at all) in the text. The ²¹⁰Pb_{ex} shows quite recent events in terms of geological time. Can you elaborate briefly in the text?

RESPONSE: We agree, the background for these measurements and how the sediment accumulation and deposition dynamics is derived from such data was not sufficiently explained. The relevant paragraph now reads (starting on line 134):

Sedimentation rates per sediment sample and per site were estimated using gamma-ray spectrometry of excess ²¹⁰Pb (²¹⁰Pb_{ex}). This analytical method utilizes decay of ²¹⁰Pb_{ex} (half-life of 22.3 years) as a tracer for sediment mixing and deposition during the past ~120 years. Vertically constant values typically indicate mixing events induced either by seismic activity or bioturbation, while exponential decline in deeper sediment layers typically reflect stable and constant deposition rates. The insight provides important information on pathways and chronology for deposited sediment layers. More specific details on material deposition and organic carbon turnover in the Atacama Trench region can be found elsewhere.

A4: “Observed PCB concentrations in surface sediment (upper 20 mm) varied”

R4: You write earlier in the text: “Five sediment layers, of 20 mm each, were analyzed from each site”. Figure 3 show 0-10, and then 10-30 mm 30-50 mm (up to 90 mm with 20 mm sediment intervals). So, this (upper 20 mm) needs to be 0-20 mm, and probably explained better. Also, replace “varied” with ranged.

RESPONSE:

All figures have been revised, for improved visuality and clarity.

If not mistaken, you overall have 3 or 4 out of 7 PCB congeners used by ICES. So, it might be beneficial to give the general overview that you identified 6 PCB congeners with concentrations ranging from ~1.5 pg g⁻¹ to 14(?) pg g⁻¹. And then continue by explaining that you detected three/four out of the 7 marker PCBs used by the International Council for the Exploration of the Sea (ICES) to determine degree of PCB contamination in the sea (ref). And then focus on PCB110 PCB180 and explain why you chose these 2 specifically.

RESPONSE:

Thank you for this comment. We have followed the reviewer’s advice. The revised sentence reads (starting on line 166):

“Observed concentrations of nine PCB congeners in surface sediment (upper 20 mm) ranged between 0.72 and 41.2 pg g⁻¹ dwt. Four of the seven indicator PCBs used by the International Council for the Exploration of the Sea (ICES) were found (#118, #138, #153, #180), and two of these (PCB#118 and PCB#180) environmentally ubiquitous PCBs are highlighted here for comparison with other studies (data on all congeners available in Figure 2 and Table S2). Concentrations of PCB#118 ranged between 1.5 and 2.5 pg g⁻¹ dwt and PCB#180 between 1.8 and 5.5 pg g⁻¹ dwt.”

A5: These concentrations correspond to, or are lower than, the reported concentrations in Arctic Ocean shelf surface (0-10 mm; 0-20 mm or 0-50 mm) sediments^{28,29}, while being several orders of magnitude lower than concentrations found in areas known to be heavily impacted by anthropogenic activities, such as the Baltic Sea^{30,31}. Yet the findings in some of the deepest and presumed most isolated regions of the global ocean is noteworthy.

R5: This is a long sentence; I have edited it and broke it in two parts. “These concentrations correspond to, or are lower than those reported from surficial sediments collected from Arctic shelf seas and Arctic interior basins, (0 up to 50 mm; XX -ZZ ng g⁻¹)^{28,29}. Nonetheless, the detected PCBs are XX-fold lower when compared to those measured from heavily impacted marine settings like the Baltic Sea^{30,31}. Yet, the findings in some of the deepest and presumed most isolated regions of the global ocean is noteworthy.

RESPONSE:

This text has been revised in accordance with suggestions by the reviewer (starting on line 176):

“The PCB concentrations reported here are lower than those reported from surficial sediments collected from Arctic shelf seas, (0 up to 50 mm; PCB#118 2-64 pg g⁻¹ dwt; PCB#180 1-41 pg g⁻¹ dwt). PCBs were mainly produced and used in the North, between 30°N and 60°N. Emissions of PCBs therefore mainly occurred in the Northern Hemisphere. These compounds are semi-volatile and can be transported through the atmosphere, by rivers and ocean currents. Through the so called grasshopping and cold condensation effects, PCBs are transported to the Arctic, and to a high extent kept there due to the influence of low temperatures on the mobility of these chemicals. Therefore, concentrations of PCBs can be expected to be higher in the Arctic compared to in remote areas in

the Southern Hemisphere. Deng et al. analyzed PCBs in sediment collected in the Antarctica (154-3240 m depth) and the South China Sea (1380-4000 m depth). PCB#180 was not included in that study, but PCB#118 (co-eluting with PCB#106 and #108) had reported concentrations of 160-280 pg g⁻¹ dwt in the Antarctica and 20-40 pg g⁻¹ dwt in the South China Sea, thus 1-2 orders of magnitude higher than our observations in the Atacama Trench. The detected PCBs are 300-1500 fold lower when compared to those measured from heavily impacted marine settings like the Baltic Sea, which contrasts to previous observations of PCB concentrations in hadal amphipods. Yet, the findings of persistent, toxic and bioaccumulating man-made chemicals in some of the deepest and presumed most isolated regions of the global ocean is noteworthy.”

I understand the comparison to other marine environments, especially if they are remote (regardless if hadal or not), but the Baltic? Anything compared to Baltic is less. The authors have extensively worked in PCBs. Nonetheless, this reviewer believes that including more relative comparisons with whatever deep-sea data are available, even if data are sketchy (e.g., Dasgupta et al 2018), needs to be made. The potential caveats of such comparisons can be pointed out in the text in a sentence.

(e.g., <https://www.sciencedirect.com/science/article/pii/S0025326X15000648>)

RESPONSE:

We thoroughly revised this paragraph and provided more comparisons to other deep-sea regions. The comparison to the Baltic Sea was there to demonstrate that although earlier reports of PCB concentrations in amphipods were comparable to heavily impacted areas (such as the Baltic), our findings of PCBs in Atacama sediment were not. This has been clarified in the text (starting on line 171):

“Concentrations of PCB#118 ranged between 1.5 and 2.5 pg g⁻¹ dwt and PCB#180 between 1.8 and 5.5 pg g⁻¹ dwt. PCB concentrations in the Atacama sediments are comparable to or lower than those reported earlier from the Mariana trench (PCB#118 0-13 pg g⁻¹ dwt; PCB#180 0-19.3 pg g⁻¹ dwt). Concentrations of several other PCB congeners reported from the Mariana trench were surprisingly high, and the authors of that study concluded that concentrations were “far higher than those recorded before in marine sediments from shallower depths”. The PCB concentrations reported here are lower than those reported from surficial sediments collected from Arctic shelf seas, (0 up to 50 mm; PCB#118 2-64 pg g⁻¹ dwt; PCB#180 1-41 pg g⁻¹ dwt). PCBs were mainly produced and used in the North, between 30°N and 60°N. Emissions of PCBs therefore mainly occurred in the Northern Hemisphere. These compounds are semi-volatile and can be transported through the atmosphere, by rivers and ocean currents. Through the so called grasshopping and cold condensation effects, PCBs are transported to the Arctic, and to a high extent kept there due to the influence of low temperatures on the mobility of these chemicals³³. Therefore, concentrations of PCBs can be expected to be higher in the Arctic compared to in remote areas in the Southern Hemisphere. Deng et al. analyzed PCBs in sediment collected in the Antarctica (154-3240 m depth) and the South China Sea (1380-4000 m depth). PCB#180 was not included in that study, but PCB#118 (co-eluting with PCB#106 and #108) had reported concentrations of 160-280 pg g⁻¹ dwt in the Antarctica and 20-40 pg g⁻¹ dwt in the South China Sea, thus 1-2 orders of magnitude higher than our observations in the Atacama trench. The detected PCBs are 300-1500 fold lower when compared to those measured from heavily impacted marine settings like the Baltic Sea, which contrasts to previous observations of PCB concentrations in hadal amphipods. Yet, the findings of persistent, toxic and bioaccumulating man-made chemicals in some of the deepest and presumed most isolated regions of the global ocean is noteworthy.”

“Yet, the findings in some of the deepest and presumed most isolated regions of the global ocean is noteworthy.”

Well, this is your hot finding. You need to put it upfront already from your abstract. For this reviewer the fact that you present the first data on PCBs is important, but not as important as that PCBs are found in geographically remote and isolated ocean areas, like trenches.

RESPONSE:

Thank you for this comment, we followed the reviewer’s advice and added this comment to the abstract (line 29-30):

“The finding of hazardous man-made chemicals in some of the deepest and presumed most isolated regions of the global ocean is noteworthy.”

A6: Figure 2.

R6: Better to reorganize Figure 2 with depth (shallow to deeper stations) instead of site numbers (1-9). Also increase scale bar to accommodate Atacama 4 (unless you have a larger scale and is not clearly shown on my copy). If levels on At4 are high, and visually undermine the other concentrations, then either use // and “break” the At 4 153, or make it a multi-panel figure.

RESPONSE:

We rearranged Figure 2 accordingly with depth, as suggested by the reviewer, and included all nine PCB congeners. The scale on the y-axis was revised to accommodate PCB153 in Atacama 4.

Also, I would suggest the authors to search for statistically significant differences between congeners and between stations, and comment appropriately.

RESPONSE:

Due to the unique nature of these samples, we did not have enough material for replicate analyses. We therefore do not have the basis for statistical analyses of the data.

A7: “The surface sediment at the two shallowest sites (Atacama 1, Atacama 9) is most certainly affected by bioturbation.”

R7: The way written is confusing. You write: “is most certainly affected by bioturbation”. The legend of Photo S1 from the whole core says: “Distinct laminated structures in deeper layers suggest absence of bioturbation by infauna and occasional deposition events.” Please, elaborate. Also, if you find bioturbation in the surface it would be beneficial to point it with an arrow or something in the photo.

Now, If bioturbation is prominent why it should be restricted only in the shallow sites? Evidence of organisms that bioturbate exists at deeper depths of other hadal trenches (Puerto Rico trench), albeit at lower abundancies. So someone could argue that biorurbation can occur at deeper hadal depths but at lower rates due to low abundancies of bioturbators.

RESPONSE:

The surface sediment at the two shallow sites appeared to be affected by bioturbation (Oguri et al 2022). During the expedition more than 50 sediment cores were recovered along the trench axis and no infauna capable of bioturbation was encountered, this is also evident from the distinct sediment layering (e.g. Fig S1, Oguri et al 2022). However, we cannot exclude that surface sediment was mixed by epifauna such as holothurians and scavenging amphipods. In addition, seismic activity has been

shown to enable mixing of sediment surfaces through trembling (Oguri et al 2016). The text was revised to (starting on line 207):

“The surface sediment at the two shallowest sites (Atacama 1, Atacama 9) may be affected by bioturbation (see Figure S1 for photo of sediment core). While larger bioturbating infauna in hadal sediment were rare, as evidenced by the distinct layering in Figure S1, epifauna such as scavenging amphipods and holothurians could still contribute to mixing of the surface layers.”

A8: mass wasting events

R8: Define for the reader (e.g., XX).

RESPONSE:

We acknowledge that a definition is required. The following paragraph has now been included (starting on line 67):

“This process is facilitated by tidal induced internal seiche and downslope gravitational driven sediment displacement. Apart from the more continuous downslope material transport, distinct sudden translocation of large amounts of previously deposited material, typically induced by earthquakes, may occur along the trench axis. Detailed investigation of $^{210}\text{Pb}_{\text{ex}}$ profiles have shown that such “mass wasting” events contribute significantly to material deposition in the Atacama trench.”

A9: “slopes to the trench axis20,32”

R9: This is true. Nonetheless, differences of organic matter occur also between the different sediment interiors, independent of slope or bottom axis sites, and could be due to different available nutrient pools (e.g. NO_x , NH_4) and presence/absence of O_2 . All these parameters control the microbial community structure, and thus how organic matter is utilized/disproportionate. So please elaborate.

RESPONSE:

These processes are complex and not the focus of this study. For some brief information, the following was added to the manuscript (line 142-145):

“Briefly, benthic mineralization was dominated by aerobic mineralization and bottom water concentrations of O_2 and NO_3^- were similar along the trench axis. Further, the O_2 penetration depth was generally similar across sites, ranging between 3.1-4.1 cm along the trench axis.”

The early diagenesis of organic material along the trench axis was investigated in detail by Glud et al 2021, Thamdrup et al 2021, and a further study on anaerobic diagenesis is in the pipeline (Glud et al in prog). In addition, a paper that in detail assesses the microbial community structure has been published (Schauberger et al 2021).

A10: The detection of PCBs in the deepest analyzed sediment layers at several sites is a sign of sediment mixing.

R10: Vaguely written. Also, this is evident from what?

RESPONSE:

The sentence was revised and now reads (line 216-218):

“The detection of PCBs in the deepest analyzed sediment layers at several sites is a sign of sediment mixing, as PCBs cannot be present in sediment deposited decades before the onset of industrial production of PCBs (in the 1930s).”

A11: with some care

R11: Consider “with caution” or “overinterpretation” as terms.

RESPONSE:

We changed to “with caution”.

A12: “Yet, our observations of increasing concentrations towards surface sediment layers contrast to the commonly reported peak concentrations of POPs in sediment layers deposited during the 1980s-1990s^{30,33}, following upon significantly reduced global emissions of PCBs in the mid-1970’s and a stop of global production of PCBs in the 1990s (Figure 3).”

R12: The reader needs clearly to understand why this is important. The data sets you are referring to are from organic-rich coastal sediments, and sediments from offshore marine settings and harbors. These areas have a history of pollution and accumulation of PCBs. So, rephrase to indicate the strength of your contradictory finding, and provide potential explanation. Also, this is a long sentence. Break it in 2.

RESPONSE:

The paragraph was revised in response to the reviewer. It now reads (starting on line 219):

“Yet, our observations of increasing concentrations towards surface sediment layers contrast to the commonly reported peak concentrations of POPs in sediment layers deposited during the 1980s-1990s. If confirmed in future studies, the observation of generally increasing PCB concentrations towards the sediment surface may indicate that sediment concentrations have not yet peaked in this remote area, with important implications for dispersal time for POP deposition in trenches. For PCBs, global emissions were significantly reduced in the mid-1970’s, followed by a stop of global production of PCBs in the 1990s (Figure 2).”

The following sentence was added to the end of the section (line 245-248):

“Future studies are needed to further investigate the concentrations of PCBs and other legacy POPs in sediment cores from the deep sea. If POPs in deep sea sediment have not yet reached peak concentrations, it poses questions about the time scale of global transport of POPs.”

A13: Figure 3.

R13: Explain why you picked PCB138 (be consistent with #). Replace: “Missing data means the concentration was below detection limit” with “Missing data indicate PCB#138 concentrations below the detection limit.”

RESPONSE:

The figures have been revised, and all nine measured PCB congeners included.

Also, Atacama 2 and Atacama 4 are X km apart, they are collected from approximately same water

depths, have same levels of TOC, they present their peaks at 50 mm, their 210Pbex profiles are similar (max difference is 36 yrs which in chronological ages can be considered marginal); yet one site has measurable PCB#138 while the other does not. Now, if you compare these findings with those from the third deeper site (Atacama 6) you see differences on the layer where you find the peak, and also on the 210Pbex profile. Can you discuss more these differences?

RESPONSE:

The explanation of why there was no PCB#138 detected in Atacama 4 is that the peak did not meet the qualification criteria. The sediment samples were run on the GCMSMS in different batches, and the batch with Atacama 4.3 had higher blank levels than the other batches.

A14: In the shallow (average 52 m, maximum 459 m) and contaminated Baltic Sea

R14: In the shallow and contaminated Baltic Sea (water depth \leq 459 m),

RESPONSE:

Ok, change done.

A15: "Secondary sources such as release from"

R15: Potential explanations include secondary sources....

RESPONSE:

Ok

A16: occurring in the late 1990's to early 2000s, were not detected in the subsurface Atacama sediment layers.

R16: Can you put this in under a general context? E.g., 1) changes in legislation that helped in PCB reduction, 2) geographically remoteness of Atacama, 3) you sampled deep-sea and hadal depths so depositions/accumulations of PCBs, can be less profound because of 1) and 2)? Just a suggestion.

RESPONSE:

This sentence referred to the possibility that we potentially did not detect peak concentrations in our 20 mm sediment samples because of the low time resolution. With thinner sediment slices, representing shorter time intervals, it might have been possible. We revised the sentence, which now reads (line 241-243):

"It is therefore possible that peak concentrations occurring in the late 1990's to early 2000s, were not detected in the subsurface Atacama sediment layers due to low time resolution of the 20 mm samples."

Organic matter turnover as a driver for PCB sediment concentrations

R1: This section is nicely written. Some overall comments to be considered:

1. You are referring to organic matter degradation by microbes. Right? If yes, then please comment briefly (sentence or two) about energy processes/microbial communities of Atacama.

RESPONSE:

Yes, we refer to microbial degradation. We added the following text to an earlier part of the manuscript (Line 142-145):

“Briefly, benthic mineralization was dominated by aerobic mineralization and bottom water concentrations of O₂ and NO₃⁻ were similar along the trench axis. Further, the O₂ penetration depth was generally similar, ranging between 3.1-4.1 cm along the trench axis.”

2. You make a valid comparison between PCBs and PAHs. Depending on the degree of aromaticity, presence of oxygen, redox conditions etc, PAHs can be degraded by microbes. Hadal studies have shown potential for microbial hydrocarbon degradation (e.g., Mariana trench). So, does the suggested release of PCBs upon organic matter degradation, could also imply that fraction of these PCBs can be utilized by hadal microbes? Microbial diversity in trenches is quite large and novel species exist with unknown metabolic potentials. You might want to point in your text something similar.

RESPONSE: Thank you for this comment. We observed indications of anaerobic dechlorination of PCBs in some of our hadal sediment samples. However, data including a broader range of PCB congeners, and sequencing of potential dehalogenating genes would be needed to take this further. We added a comment on this to the manuscript (line 314-316):

“This limitation also prevented further investigations of the possibility of anaerobic dechlorination of highly chlorinated PCB congeners in hadal sediment samples.”

3. Would you consider an overall schematic representation of what you suggest?

RESPONSE:

With six multi-panel figures in the revised manuscript, we believe there is no room for an additional one. If allowed by the journal, we would still be happy to produce one.

A2: “Sediment concentrations of PCBs”

R2: Rephrase to: Sediment concentrations of three marker PCBs

RESPONSE:

Ok

A3: expressed on an organic-carbon basis (pg g⁻¹ OC).

R3: For this normalization did you use your TOC data? How did you deal with the sediment depths below 10 mm?

RESPONSE:

Correct, we normalized to TOC. We have added the TOC data in all sediment samples to Table S1 and refer to that in the text (line 133).

Reviewer #1 (Remarks to the Author):

The authors have adequately addressed the comments raised.

Reviewer #2 (Remarks to the Author):

The manuscript has been improved and reads really well.
This reviewer endorses this study for publication.